# A systems-level study reveals host-targeted repurposable drugs against SARS-CoV-2 infection

Fangyuan Chen[1,2,†] (iD), Qingya Shi[1,2,†] (iD), Fen Pei[1,3,†] (iD), Andreas Vogt[1,3,†] (iD), Rebecca A Porritt[4,5], Gustavo Garcia Jr[6,7], Angela C Gomez[4], Mary Hongying Cheng[1], Mark E Schurdak[1,3], Bing Liu[1] (iD), Stephen Y Chan[8,9], Vaithilingaraja Arumugaswami[6,7], Andrew M Stern[1,3] (iD), D Lansing Taylor[1,3], Moshe Arditi[4,5] (iD) & Ivet Bahar[1,3,*] (iD)

## Abstract

Understanding the mechanism of SARS-CoV-2 infection and identifying potential therapeutics are global imperatives. Using a quantitative systems pharmacology approach, we identified a set of repurposable and investigational drugs as potential therapeutics against COVID-19. These were deduced from the gene expression signature of SARS-CoV-2-infected A549 cells screened against Connectivity Map and prioritized by network proximity analysis with respect to disease modules in the viral–host interactome. We also identified immuno-modulating compounds aiming at suppressing hyperinflammatory responses in severe COVID-19 patients, based on the transcriptome of ACE2-overexpressing A549 cells. Experiments with Vero-E6 cells infected by SARS-CoV-2, as well as independent syncytia formation assays for probing ACE2/SARS-CoV-2 spike protein-mediated cell fusion using HEK293T and Calu-3 cells, showed that several predicted compounds had inhibitory activities. Among them, salmeterol, rottlerin, and mTOR inhibitors exhibited antiviral activities in Vero-E6 cells; imipramine, linsitinib, hexylresorcinol, ezetimibe, and brompheniramine impaired viral entry. These novel findings provide new paths for broadening the repertoire of compounds pursued as therapeutics against COVID-19.

**Keywords** autophagy; SARS-CoV-2-infected cell transcriptomics; syncytia formation; viral entry; viral–host interactions
**Subject Categories** Microbiology, Virology & Host Pathogen Interaction; Pharmacology & Drug Discovery
**Mol Syst Biol. (2021) 17: e10239**

## Introduction

Coronavirus disease-2019 (COVID-19) caused by severe acute respiratory syndrome coronavirus (CoV) type 2 virus (SARS-CoV-2) has led to over 3 million deaths as of April 2021, and there is an urgent need to better understand the mechanisms of infection and the host cell response and to develop new therapeutics. Identification of repurposable drugs became a widespread approach for addressing current pharmacological challenges, including those faced by the current pandemic. Many compounds under clinical trials against SARS-CoV-2 are potentially repurposable drugs (Esposito *et al*, 2020; Tu *et al*, 2020) that target viral proteins. While such efforts are worth pursuing, an alternative strategy is to discover host-targeted therapies. We focus here on the identification of repurposable compounds that modulate host cell responses, using a comprehensive, mechanism unbiased, and highly integrated systems-level approach.

The current quantitative systems pharmacology (Stern *et al*, 2016) approach leverages recent progress in the field in an integrated computational/experimental framework: One is the rigorous evaluation of the differentially expressed genes (DEGs) in SARS-CoV-2-infected cells, and the use of these DEG patterns for extracting from the Connectivity Map (CMap) database (Lamb *et al*, 2006; Subramanian *et al*, 2017) candidate compounds/drugs that would reverse the infected cells' transcriptional program. Recent study showed, for example, the success of a CMap-based drug signature refinement approach for improving drug repositioning predictions (Iorio *et al*, 2015). Here, we use the transcriptome data from SARS-CoV-2-infected A549 (human adenocarcinomic alveolar basal epithelial) cells (preprint: Blanco-Melo *et al*, 2020a) from lung tissue, as well as those of A549 cells overexpressing the host cell

1   Department of Computational and Systems Biology, School of Medicine, University of Pittsburgh, Pittsburgh, PA, USA
2   School of Medicine, Tsinghua University, Beijing, China
3   University of Pittsburgh Drug Discovery Institute, Pittsburgh, PA, USA
4   Department of Pediatrics, Division of Pediatric Infectious Diseases and Immunology, Cedars-Sinai Medical Center, Los Angeles, CA, USA
5   Biomedical Sciences, Infectious and Immunologic Diseases Research Center, Cedars-Sinai Medical Center, Los Angeles, CA, USA
6   Department of Molecular and Medical Pharmacology, David Geffen School of Medicine, University of California, Los Angeles, CA, USA
7   Eli and Edythe Broad Center of Regenerative Medicine and Stem Cell Research, University of California, Los Angeles, CA, USA
8   Pittsburgh Heart, Lung, Blood, and Vascular Medicine Institute, University of Pittsburgh Medical Center, Pittsburgh, PA, USA
9   Division of Cardiology, Department of Medicine, University of Pittsburgh Medical Center, Pittsburgh, PA, USA
    *Corresponding author. Tel: +1 412 648 3332; E-mail: bahar@pitt.edu
    †These authors contributed equally to this work

receptor angiotensin-converting enzyme 2 (ACE2) (Blanco-Melo *et al*, 2020b). The latter ensures high multiplicity of infection and allows for observing the DEGs under severe infection.

Another important advance is the characterization of virus–host cell interactome for SARS-CoV-2 (Gordon *et al*, 2020b) and knowledge of cell-specific protein–protein interaction (PPI) networks. These data, combined with network-based proximity analysis (Guney *et al*, 2016), may help quantify the extent of interaction between the targets of each compound and the host cell proteins participating in the interactome with the virus. For example, Zhou *et al* (2020b) recently proposed 16 repurposable drugs using a network proximity analysis between drug targets in the human PPIs and host cell proteins associated with four human CoVs (SARS-CoV, MERS-CoV, HCoV-229E, and HCoV-NL63), the mouse MHV, and avian IBV, but not SARS-CoV-2.

We also have access to increasingly larger databases on protein-target interactions and target-pathway mappings, and interfaces such as our QuartataWeb webserver (Li *et al*, 2020) that permit to identify and/or predict drug-target associations and to bridge targets to cellular pathways completing chemical-target-pathway mappings.

We report the identification of 15 compounds, including repurposable and investigational drugs, that are proposed to act against SARS-CoV-2 upon targeting the host cell machinery. *In vitro* assays conducted in Vero-E6 cells, HEK293T cells, and Calu-3 lung cancer cells for 10 of these prioritized compounds—six repurposable FDA-approved drugs (imipramine, salmeterol, hexylresorcinol, brompheniramine, ezetimibe, and temsirolimus) and four under development (linsitinib, torin-1, rottlerin, semaxanib)—demonstrated that several of them inhibited SARS-CoV-2 viral entry in a dose-dependent manner, with linsitinib being particularly effective. Additionally, we propose 23 compounds for possible anti-hyperinflammatory (adjuvant) actions. These findings expand the repertoire of drugs/compounds that could be repurposed/developed for possible COVID-19 treatment.

# Results

### Overall workflow

Figure 1 schematically describes the computational workflow adopted in the present study. As input, we used the RNA-seq data from SARS-CoV-2-infected A549 cells (preprint: Blanco-Melo *et al*, 2020a) (referred to as Dataset 1), and those from SARS-CoV-2-infected A549 cells overexpressing ACE2 (shortly designated as A549-ACE2 cells) (Blanco-Melo *et al*, 2020b, Data ref: tenOever & Blanco-Melo, 2020, referred to as Dataset 2). We analyzed the corresponding DEGs to construct antiviral and immuno-modulating (anti-inflammatory) gene signatures respectively, which were then used to predict optimal compounds/drugs that match those signatures using CMap (Fig 1A–D). Of note, the simple signature reversal approach, as utilized in many CMap studies and a recent study of SARS-CoV-2 (preprint: Duarte *et al*, 2020) is not applicable here, because part of the infection-induced signature promotes viral life cycle while another part reflects antiviral responses which should be promoted rather than suppressed. To address this point, we have selected 36 and 17 DEGs from the two respective datasets, whose actions should be either reversed or promoted by CMap-deduced

drugs/compounds, depending on their role in the host proteome, as will be presented in the next subsection.

Following the identification of the compounds or repurposable drugs expected to reverse the SARS-CoV-2 pathogenic (and not the host cell immunoprotective) effects (Fig 1D), we prioritized a subset following the network proximity analysis introduced by Guney *et al* (2016) (Fig 1E–G). To this aim, we used the SARS-CoV-2-host interactome (Gordon *et al*, 2020b) and the lung PPI network in the BioSNAP dataset (Zitnik *et al*, 2018) (Fig 1F). We first identified four *disease modules*—viral entry, viral replication and translation, cell signaling and regulation, and immune response modules in the viral–host interactome; and then, we evaluated the "distance" of each compound from each disease module based on the proximity of the compounds' targets to the proteins belonging to the module using the lung PPI network in BioSNAP (Fig 1G).

The compounds "closest" to each module, called the prioritized compounds, were then analyzed and clustered based on their interaction patterns with targets using QuartataWeb (Li *et al*, 2020), to select representatives from each cluster (Fig 1H). Additional criteria, such as drug development status, side effects, mechanism of action (MOA), and antiviral activities from databases and/or literature, were considered in making the final selections from among the cluster representatives for experimental tests and possible validation (Fig 1I). We provide below more specifics on the successive steps and outputs.

### Antiviral and anti-inflammatory signatures derived from post-SARS-CoV-2 infection transcriptomics

We identified 120 DEGs composed of 100 upregulated and 20 downregulated genes by DESeq2 analysis (Love *et al*, 2014) of the transcriptome of SARS-CoV-2-infected A549 cells (Dataset 1), using false-discovery rate (FDR) default upper value of 0.05 (Fig 2 A and Appendix Table S1). Gene Ontology (GO) (Ashburner *et al*, 2000; UniProt Consortium, 2019) enrichment analysis of the 100 upregulated genes showed that they were mainly involved in viral life cycle and some in early defensive immune responses mediated by interferons (IFNs; Fig 2B). Such early responses include viral translation inhibition, RNA degradation, RNA editing, or nitric oxide synthesis (Samuel, 2001). Nevertheless, the induction of interferon types I and III was relatively more "muted" in SARS-CoV-2-infected A549 cells compared to those of other respiratory viruses such as influenza A and respiratory syncytial virus (Blanco-Melo *et al*, 2020b).

As to downregulated genes, they mainly comprised vesicle-related structures or endosomal events, including autophagosome formation for autophagic elimination of the virus (Kudchodkar & Levine, 2009). Promoting autophagy showed potential in reducing MERS infection (Gassen *et al*, 2019) and thus down-regulation of this process might contribute to viral escape. In CMap applications to diabetes (Zhang *et al*, 2015) and obesity (Liu *et al*, 2015), compounds that reverse the gene signature induced by the disease were selected. However, in SARS-CoV-2 infection, it is important to promote the adaptive immune response mediated by IFNs at early stage rather than blindly reversing the complete gene signature. Therefore, after overrepresentation analysis, and evaluation of the GO annotations associated with these genes as described in the Materials and Methods, we selected 36 genes to be upregulated

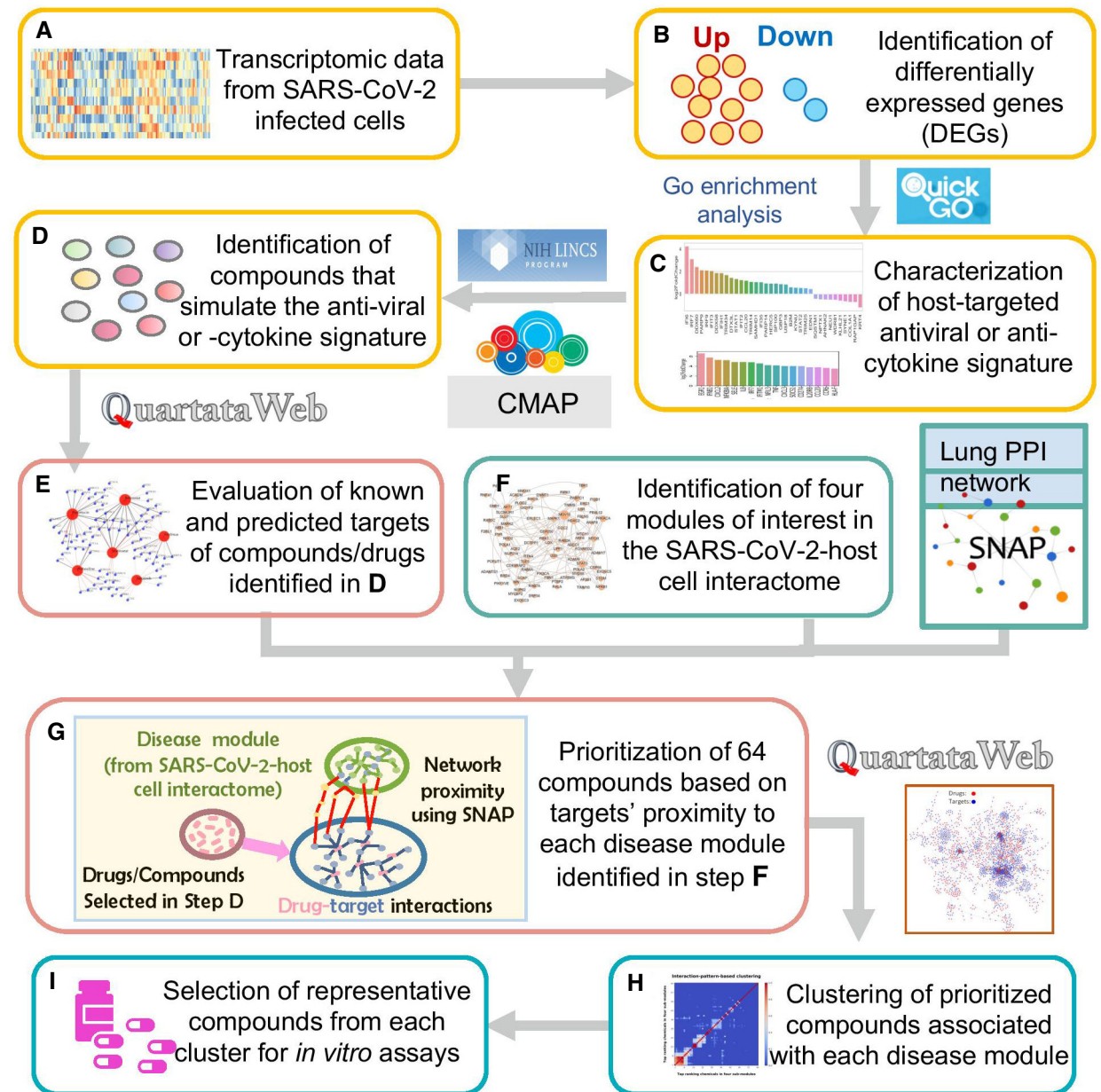

**Figure 1. Workflow of the quantitative systems pharmacology approach for selecting compounds for experimental evaluation.**

A  We use as input the RNA-seq data from SARS-CoV-2 infected A549 cells (preprint: Blanco-Melo *et al*, 2020a) and ACE2-overexpressing A549 cells (Blanco-Melo *et al*, 2020b).

B  Up- and downregulated differentially expressed genes (DEGs) were identified from these data using Wald test with false-discovery rate (FDR) default upper value of 0.05.

C  The antiviral gene signature (top) and anti-cytokine gene signature (bottom) were identified upon manual curation of GO enrichment results corresponding to the DEGs, using the QuickGO (Binns *et al*, 2009) hierarchical annotation (see Fig 2 for details).

D  Two sets of compounds or repurposable drugs that best reproduced the antiviral and anti-cytokine signatures were extracted from CMap (Lamb *et al*, 2006; Subramanian *et al*, 2017).

E  Known and predicted targets of these compounds were identified using QuartataWeb (Li *et al*, 2020).

F  A host response network composed of four modules related to SARS-CoV-2 infection (called *disease modules*) was constructed.

G  The target of the compounds identified in (E) and the disease modules in (F) were subjected to network proximity analysis (Guney *et al*, 2016) using BioSNAP lung PPI network, to prioritize 25 repurposable or investigational drugs for each module. This step has been performed for antiviral compounds only.

H, I  The compounds were clustered based on the interaction patterns with their targets, using QuartataWeb. Representatives from each cluster (H) and additional compounds identified by manual curation were selected for experimental testing (I).

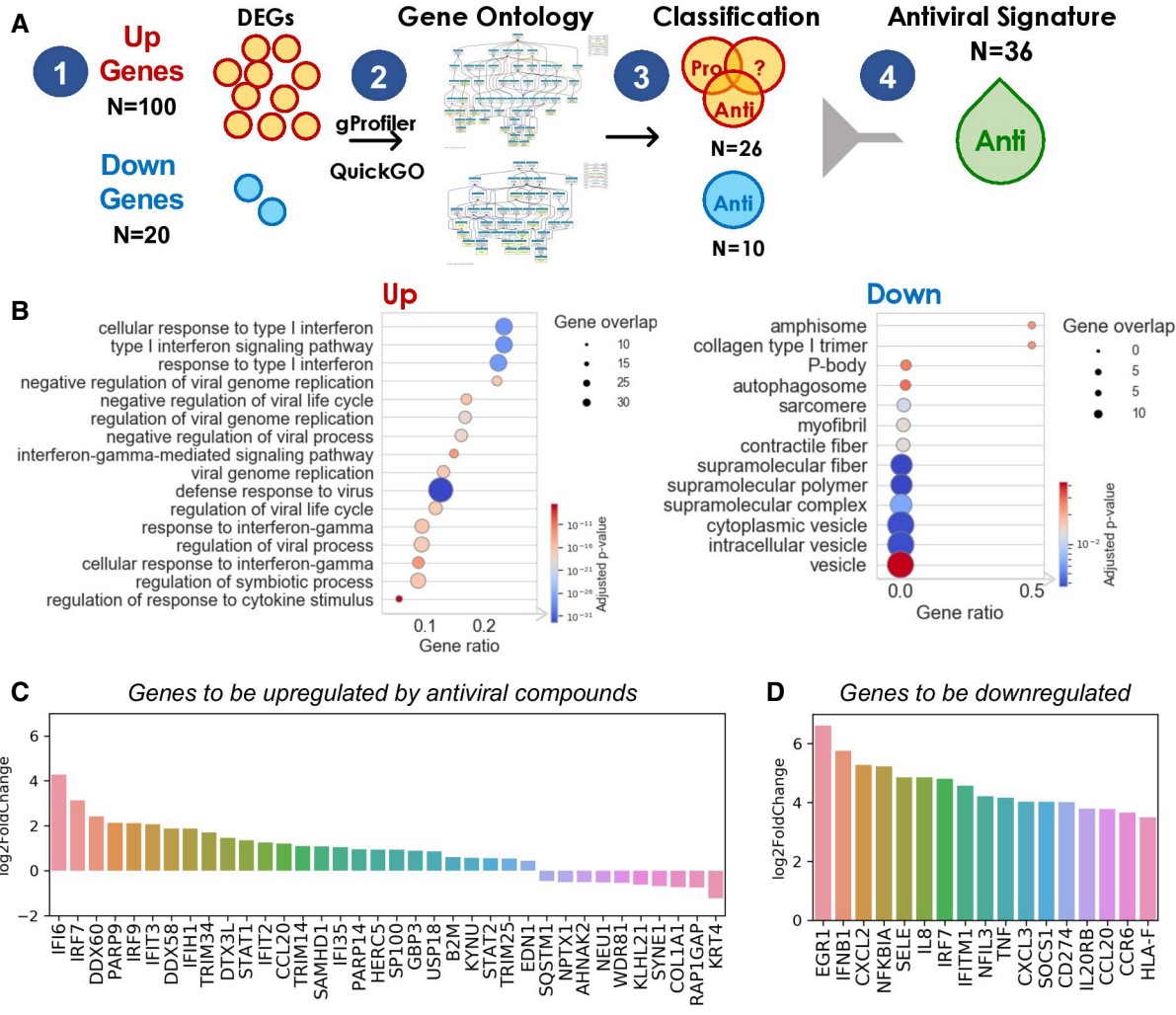

**Figure 2. Antiviral and anti-cytokine signature derived from the post-SARS-CoV-2-infection transcriptome the respective A549 and A549-ACE2.**

A   Illustration of the *4-step* pipeline for identifying the intrinsic antiviral signature in A549 cells 24 h after SARS-CoV-2 infection: (1) Identification of 100 upregulated and 20 downregulated genes; (2) GO enrichment analysis for up- and downregulated genes, respectively. The hierarchy of enriched GO terms was generated using QuickGO; (3) Classification of pro- or antiviral GO terms. Upregulated GO terms are classified as either proviral, antiviral or ambiguous. Downregulated GO terms are all considered as anti-viral; (4) Gene selection for antiviral signature from the classified GO terms. Genes were included if they were antiviral or unknown.

B   GO enrichment of up (left) and down (right) regulated genes. GO terms were filtered by size and overlapping genes as described in Materials and Methods. A total of 17 upregulated (Biological Process) and 13 downregulated (Cellular Component) genes are illustrated. *P*-values were derived from Fisher's one-tailed test and adjusted by Benjamini–Hochberg for multiple test correction.

C   Change in the expression levels of 36 genes defining the host-targeted antiviral signature. $\log_2$ fold change at 24-h post-SARS-CoV-2 infection from A549 cells are shown.

D   Change in the expression levels of 17 genes defining anti-cytokine signature; $\log_2$ fold change at 24 h post-high SARS-CoV-2 infection from A549-ACE2 cells are shown.

(Fig 2C). These genes are comprised of (i) 26 genes upregulated in SARS-CoV-2 infected A549 cells, which are associated with viral defense and should be upregulated for antiviral activity, and (ii) 10 genes downregulated in A549 cells, associated with endocytic or vesicular processes, which should be reverted. Table 1 lists the corresponding gene products/proteins (left two columns). Appendix Table S2A provides information on their GO biological processes.

The A549-ACE2 cells (Dataset 2) repeatedly exhibited a more pronounced cytokine upregulation, along with IFN response insufficiency, compared to A549 cells. Based on this observation, immune-modulating therapies have been suggested (Blanco-Melo *et al*, 2020b). We selected the most strongly upregulated 17 genes ($\log_2$fold change of 3.5 or higher; see Materials and Methods), toward identifying compounds that would suppress the excessive inflammatory cytokine response in severe COVID-19 patients. This led to the anti-inflammatory (or anti-cytokine) signature shown in Fig 2D composed of 17 genes to be downregulated (Table 1 right two columns). Appendix Table S2B list the corresponding proteins and their GO annotations.

**Table 1.   Antiviral and anti-inflammatory signature genes derived from SARS-CoV-2-infected cells.**

| Antiviral signature (based on A549 Cells) | | Anti-inflammatory signature (A549-ACE2 cells) | |
|---|---|---|---|
| **Gene[a]** | **Protein[b]** | **Gene[c]** | **Protein[b]** |
| **To-be-upregulated** | | **To-be-downregulated** | |
| *IFI6* | IFNα-inducible protein 6 | *EGR1* | Early growth response protein |
| *IRF7* | IFN regulatory factor 7, isoform CRA_a | *IFNB1* | Interferon beta |
| *DDX60* | ATP-dependent RNA helicase DDX60 | *CXCL2* | C-X-C motif chemokine |
| *PARP9* | Protein mono-ADP-ribosyltransferase PARP9 | *NFKBIA* | NFκB inhibitor α |
| *IRF9* | IFN regulatory factor 9 | *SELE* | E-selectin |
| *IFIT3* | IFN-induced protein with tetratricopeptide repeats 3 (Retinoic acid-induced gene G protein) (RIG-G) | *IL8* | Interleukin-8 |
| *DDX58* | Antiviral innate immune response receptor RIG-I | *IRF7* | IFN regulatory factor 7 |
| *IFIH1* | IFN-induced helicase C domain-containing protein 1 | *IFITM1* | IFN-induced transmembrane protein 1 |
| *TRIM34* | Tripartite motif-containing protein 34 (IFN-responsive finger protein 1) (RING finger protein 21) | *NFIL3* | Nuclear factor interleukin-3-regulated prot |
| *DTX3L* | E3 ubiquitin-protein ligase DTX3L (EC 2.3.2.27) | *TNF* | Tumor necrosis factor α |
| *STAT1* | Signal transducer and activator of transcription 1 | *CXCL3* | C-X-C motif chemokine 3 |
| *IFIT2* | IFN-induced protein with tetratricopeptide repeats 2 | *SOCS1* | Suppressor of cytokine signaling 1 |
| *CCL20* | C-C motif chemokine 20 (Fragment) | *CD274* | Programmed cell death 1 ligand 1 |
| *TRIM14* | Tripartite motif-containing 14, isoform CRA_c | *IL20RB* | Interleukin-20 receptor subunit β |
| *SAMHD1* | Deoxynucleoside triphosphate triphosphohydrolase | *CCL20* | C-C motif chemokine 20 |
| *IFI35* | IFN-induced 35 kDa protein (IFP 35) (Ifi-35) | *CCR6* | C-C chemokine receptor type 6 |
| *PARP14* | Protein mono-ADP-ribosyltransferase PARP14 | *HLA-F* | Human leukocyte antigen F |
| *HERC5* | E3 ISG15--protein ligase HERC5 (Fragment) | | |
| *SP100* | Nuclear autoantigen Sp-100 (Fragment) | | |
| *GBP3* | Guanylate-binding protein 3 | | |
| *USP18* | Ubl carboxyl-terminal hydrolase 18 | | |
| *B2M* | β2-microglobulin | | |
| *KYNU* | Kynureninase (Fragment) | | |
| *STAT2* | Signal transducer and activator of transcription 2 | | |
| *TRIM25* | E3 ubiquitin/ISG15 ligase TRIM25 | | |
| *EDN1* | Endothelin-1 (Preproendothelin-1) (PPET1) | | |
| **Gene[d]** | | | **Protein[b]** |
| **To-be-upregulated** | | | |
| SQSTM1 | | | Sequestosome-1 |
| AHNAK2 | | | Protein AHNAK2 |
| NPTX1 | | | Neuronal pentraxin-1 (NP1) |
| NEU1 | | | Sialidase-1 |
| WDR81 | | | WD repeat-containing protein 81 |
| KLHL21 | | | Kelch-like protein 21 |
| SYNE1 | | | Nesprin-1 |
| COL1A1 | | | Collagen, type I, α1, isoform CRA_a |
| RAP1GAP | | | Rap1 GTPase-activating protein 1 |
| KRT4 | | | Keratin, type II cytoskeletal 4 |

[a]Genes observed to be upregulated in the transcriptome of A549 cells (Dataset 1).
[b]Protein: gene product from UniProt Consortium (2019).
[c]Genes observed to be upregulated in the transcriptome of ACE2-overexpressing A549 cells (Dataset 2); All genes are ordered by log$_2$fold change in descending order. See Appendix Tables S1A and B and S2A and B for the log$_2$foldchange values and associated GO biological processes or cellular components. See also Fig 2C and D for the respective log$_2$fold change profiles observed in SARS-CoV-2-infected-A549 and SARS-CoV-2-infected-A549-ACE2 cells.
[d]genes observed to be downregulated in the transcriptome of A549 cells (Dataset 1).

## Identification of antiviral and anti-cytokine compounds and corresponding targets

The compounds that best matched the antiviral and anti-cytokine signatures determined above were identified by screening each signature against the CMap database. Briefly, the Touchstone collection of perturbagen signatures from 3,000 compounds on six cell lines was searched to assign a CMap connectivity score to each compound. The score is based on the similarity between the compound-induced gene signature in CMap and the query/input signature, repeated separately for the antiviral and anti-cytokine signatures. This led to a set of 263 potentially antiviral compounds, and another of 275 potentially anti-cytokine compounds, using default thresholds in CMap (see Materials and Methods), listed in Appendix Table S3A and B, respectively. The compounds included twelve (chlorpromazine, apicidin, ribavirin, mycophenolate, entacapone, equilin, metformin, mercaptopurine, gemcitabine, mepacrine/quinacrine, daunorubicin, and valproic acid) listed in the COVID-19 drug repurposing database compiled by Excelra (Excelra, 2020).

Of these two respective sets, 168 and 163 compounds were annotated in QuartataWeb (Li *et al*, 2020), which provided us with information on the targets of these compounds using DrugBank-all (Wishart *et al*, 2018) and STITCH-experimental (Szklarczyk *et al*, 2016) data as input. The remaining compounds were "manually" analyzed based on existing literature, as schematically described in the Appendix Fig S1 for Dataset 1. Appendix Fig S2 shows that the most frequently targeted proteins by the candidate antiviral compounds were adrenergic receptor α1A (gene *ADRA1A*), serotonin receptor 2A (*HTR2A*), and histamine H1 receptor (*HRH1*). Incidentally, *ADRA1A* and *HRH1* were also among the most upregulated genes in SARS-CoV-2 infected A549 cells (preprint: Emanuel *et al*, 2020). Elevated *HRH1* can be associated with hyperinflammation (Thurmond *et al*, 2008). Serotonin receptor 2A was maximally targeted by potential anti-cytokine compounds drawing attention to the impact on neurotransmission.

## Classification of host proteins implicated in SARS-CoV-2 infection in four modules

We considered a set of 348 SARS-CoV-2-related host cell proteins composed of 332 proteins identified by Gordon *et al* (2020b), plus 16 reportedly involved in SARS-CoV-2 life cycle (de Lartigue *et al*, 2009; Li *et al*, 2019b; Hoffmann *et al*, 2020a; Ou *et al*, 2020).

The 332 host cell proteins were identified (Gordon *et al*, 2020b) by mass spectrometry upon expressing 26 of 29 SARS-CoV-2 proteins (non-structural proteins Nsp1–16, spike [S], envelop [E], membrane [M], nucleocapsid [N], and nine open reading frames [Orfs]), individually in HEK293T cells. Comparison of the viral–human interactomes for SARS-CoV-2, SARS-CoV, and MERS-CoV (Gordon *et al*, 2020a) revealed that 14.7% of the SARS-CoV-2 host proteins were not among those detected in SARS-CoV-1 or MERS-CoV interactomes underscoring the significance of utilizing the viral/host interactome specific to SARS-CoV-2.

The additional 16 proteins are the receptor ACE2, the proteases transmembrane protease serine 2 (TMPRSS2), cathepsin B, and cathepsin L, as well as several cell signaling and regulation proteins (interleukin 6 [IL6] receptor, myeloid differentiation primary response 88 [MyD88], MAP kinase 1, protein kinase B [AKT1], mammalian target

of rapamycin [mTOR], nuclear factor of activated T cells cytoplasmic 1 [*NFATC1*], nuclear factor κB subunit 1 [NFκB1], STAT3, ADAM metallopeptidase domain 17 [ADAM17], phosphatidylinositol 3-kinase catalytic subunit α [*PIK3CA*], phosphatidylinositol 3-phosphate 5-kinase [PIKfyve], and the two-pore channel 2 [TPC2]).

In order to better assess the involvement of these 348 host cell proteins in different phases of SARS-CoV-2 infection, we mapped them onto their KEGG pathways (243 pathways) and identified four functional modules: viral entry, viral replication and translation, host cell regulation and signaling, and immune response, based on their KEGG annotations. This led to 27, 45, 27, and 32 proteins in the respective modules (see Table 2 and Appendix Table S4). Several proteins were shared between these modules, such that their union contained 103 host proteins. For example, MAPK and PI3K-AKT-mTOR signaling pathways regulate CoV replication and translation (Zumla *et al*, 2016), in addition to mediating the immune response (Prompetchara *et al*, 2020). We also note in Table 2 some proteins distinguished in a recent CRISPR screen (Daniloski *et al*, 2021), including the Ras-related protein Rab-7A (*RAB7A*), and subunits of the ATPase vacuolar pump (*ATP6AP1* and *ATP6VIA*) and intracellular cholesterol transporter (*NPC2*).

## Prioritization of candidate compounds proposed to have antiviral effects

As a measure of the potential antiviral effect of the compounds deduced from our computational analysis, we calculated the proximity of their targets to each disease module. Specifically, we evaluated the distance between the targets of each compound, and the proteins belonging to each module using the lung-specific PPI network from BioSNAP (Zitnik *et al*, 2018) and network proximity analysis (Guney *et al*, 2016) (see Materials and Methods). Top-ranking 25 compounds were selected for each module (Fig 3A and Appendix Table S5) leading to a set of 64 distinct compounds in the union of four modules (Fig 3B). Clustering of these based on their interaction patterns with target proteins (using QuartataWeb) led to 12 clusters (Appendix Fig S3A and Table S6A) containing 48 of the compounds; the remaining 16 exhibited unique interaction patterns. Up to two representatives were selected from each cluster and further evaluated (manually) with literature-based evidence including their MOAs, side effects, availability, and antiviral evidence if any, to generate a reduced set of 13 high-priority compounds, listed in Table 3. In addition, after manual evaluation of 95 compounds that lack data in DrugBank and STITCH, two investigational compounds, rottlerin, and QL-XII-47, with respective CMap scores of 97.11 and 99.03, were added to our high-priority list (see Appendix Fig S1).

The final set of 15 compounds that are proposed to have antiviral activities (Table 3) contains eight FDA-approved (repurposable) drugs and seven under investigation. Ten of these have been tested in *in vitro* assays (indicated by asterisks in Table 3; and labeled in red in Fig 3B). Appendix Fig S4 displays the corresponding chemical structures.

## Prioritization of candidate compounds proposed to have anti-inflammatory effects

A similar interaction pattern-based clustering of the 163 compounds predicted to potentially have anti-cytokine effect (among the high

Table 2. Four modules mediating host cell response during SARS-CoV-2 infection, corresponding pathways, and proteins.[a]

| Module (# of proteins) | KEGG pathways | Gene names of the host cell proteins involved in the module |
|---|---|---|
| Viral Entry (27 proteins) | Endocytosis, lysosome pathway | SCARB1; **ATP6AP1**; **NPC2**; AP3B1; ITGB1; **RAB8A**; AP2A2; PIKFYVE; RHOA; **RAB10**; **ACE2**; AP2M1; **ATP6V1A**; RNF41; CHMP2A; **CTSB**; WASHC4; **TMPRSS2**; **RAB7A**; GLA; SPART; **CTSL**; PPT1; **ARF6**; **RAB5C**; NEU1; TPC2 |
| Viral replication & translation (45 proteins) | DNA replication, RNA transport, RNA degradation, protein processing in ER and protein export | NUP62; ERLEC1; NUP214; EIF4E2; RPL36; LMAN2; EXOSC3; NUP54; WFS1; PRIM2; SRP72; SIL1; UPF1; SELENOS; POLA1; NUP88; OS9; HYOU1; RAE1; RBX1; EXOSC2; MRPS2; NUP98; PSMD8; NGLY1; NUP58; ERO1B; EDEM3; MRPS5; PRIM1; NUP210; ELOC; SRP54; ELOB; UGGT2; EXOSC5; IMPDH2; PABPC4; EXOSC8; POLA2; SRP19; SLU7; CUL2; MOGS; PABPC1 |
| Regulation and signaling (27 proteins) | Ras signaling, autophagy, AMPK signaling, mTOR signaling, PI3K-AKT signaling, and insulin signaling | IL6R; **PIK3CA**; **RAB8A**; RALA; MTOR; **TBK1**; AKT1; **NFKB1**; GNG5; EIF4E2; PRKAR2A; **RAB2A**; MYD88; **ATP6V1A**; COL6A1; RAB14; MAPK1; RHOA; **RAB10**; PRKAR2B; ITGB1; GNB1; **ARF6**; **RAB5C**; ECSIT; **PRKACA**; NFATC1 |
| Immune response (32 proteins) | Toll-like receptor-, chemokine-, RIG-like receptor-, B cell receptor-, NF-kB-, TCR-, and HIF-1-signaling pathways | MYD88; MAPK1; STAT3; CUL2; HMOX1; ELOB; RIPK1; IL17RA; CSNK2B; MTOR; INHBE; **PRKACA**; GNB1; NLRX1; ERC1; **RHOA**; GDF15; **TBK1**; IL6R; AKT1; CSNK2A2; GNG5; NFATC1; TBKBP1; **PIK3CA**; **CTSB**; RBX1; **NFKB1**; ELOC; EIF4E2; PLAT; **ARF6** |

[a]See Appendix Table S4 for the full names of the proteins whose gene codes are listed in column 3. Genes corresponding to some key proteins targeted by the proposed compounds/drug and/or mentioned in the text are written in bold face, including: ARF6 (ADP ribosylation factor 6); ATP6AV1A (ATPase H+ transporting V1 subunit A); TBK1 (TANK-binding kinase 1); PRKACA (protein kinase CAMP-activated catalytic subunit α, or the catalytic subunit α of protein kinase A (PKA); RAB7A (Ras-related protein Rab-7A; RHOA (recombinant human RhoA); CTSL and CTSB (cathepsin L and B).

CMap-scoring 275; see Appendix Table S3B) led to 20 clusters of two or more compounds based on compound–protein interaction patterns, while 35 compounds were left as singletons (Appendix Fig S3B and Table S6B). We selected 19 high-priority compounds representative of these clusters in addition to 5 singletons. Furthermore, literature search of the remaining 112 potentially anti-inflammatory compounds for which no target data were available in DrugBank and STITCH, led to three additional candidate compounds. The resulting set of 27 potentially anti-inflammatory/cytokine compounds is presented in Table 4.

Table 4 contains 15 FDA-approved drugs and 12 compounds under investigation. Of note, two of the compounds under investigation (JAK3-Inhibitor-II and AZD-8055; in *boldface*) also belong to

the 64 top-ranking compounds based on Dataset 1; and one, mepacrine/quinacrine, is listed in the Excelra COVID-19 drug repurposing database (Excelra, 2020). Another investigational drug in the list, PCA4248, is a platelet-activating factor (PAF) receptor antagonist (Fernandez-Gallardo *et al*, 1990), and its utility against COVID-19 (e.g., for preventing coagulation or blood clots) is to be explored, as well as those of the two His receptor antagonists azelastine and chlorphenamine, identified here. Recent study draws attention to the possible repurposing of PAF receptor antagonists and His receptor antagonists against hyperinflammation and microthromboses in COVID-19 patients (Demopoulos *et al*, 2020).

Among approved drugs, pirfenidone is known to inhibit furin (Burghardt *et al*, 2007), a human protease involved in the cleavage of the viral spike glycoprotein into S1 and S2 subunits (like TMPRSS2). Spike cleavage is essential to activate the S1 fusion trimer for viral entry. Pirfenidone combined with melatonin has been pointed out to be a promising therapy for reducing cytokine storm in COVID-19 patients (Artigas *et al*, 2020). Finally, Table 4 also contains two approved cyclooxygenase inhibitors, oxaprozin and dexketoprofen, known as non-steroidal anti-inflammatory drugs (NSAIDs) (Miller, 1992; Moore & Barden, 2008).

## Testing the SARS-CoV-2 inhibitory properties of prioritized compounds in *in vitro* assays

We first selected five compounds (salmeterol, rottlerin, temsirolimus, torin-1, and ezetimibe) from the list of 15 prioritized compounds described in Table 3 for a proof of concept *in vitro* evaluation of their anti-SARS-CoV-2 potential. We used a SARS-CoV-2 infectious cell culture system (Figs 4A and EV1) where host Vero-E6 cells were pretreated with compounds for 1 h prior to SARS-CoV-2 inoculation. After 48-h post-infection, we performed immunofluorescence to assess viral infection (SARS-CoV-2 S protein; Figs 4A and EV1). Images were analyzed for spike-positive cells using the Multiwavelength Cell Scoring algorithm in MetaXpress. Representative mock and vehicle control images and their segmentation are shown in Fig 4A. Violin plots describing the distribution of the log integrated spike for each cell in the untreated and treated samples are shown in Fig 4B along with complementary pie charts indicating the percent of cells positive for spike protein (Fig 4C). In the untreated controls, a bimodal distribution of spike-positive cells was evident, indicating the presence of two infected cell populations with one expressing more spike protein per cell than the other (Fig 4B). Salmeterol at 0.1 and 1 µM reduced the median of the spike-expressing population and showed a preferential antiviral effect for the lower spike-expressing subpopulation (Fig 4B). At 10 µM, salmeterol exhibited a greater antiviral effect on the entire population, although some (~ 14%) spike-positive cells were evident (Fig 4B). Qualitatively similar results to salmeterol were obtained with rottlerin and the mTOR inhibitors, Temsirolimus, and Torin-1, although dose-limiting toxicity as evidenced by reduced cell count prevented a determination of a more complete antiviral effect on the higher-spike protein-expressing subpopulation in torin-1- and RO77-treated cells (Fig 4B). Ezetimibe reduced spike protein-expressing populations only at the highest concentration studied (25 µM), where a reduction in cell numbers was also observed.

Next, we proceeded to cell fusion assays as a proxy for ACE2/SARS-CoV-2-mediated viral entry. We focused on prioritized

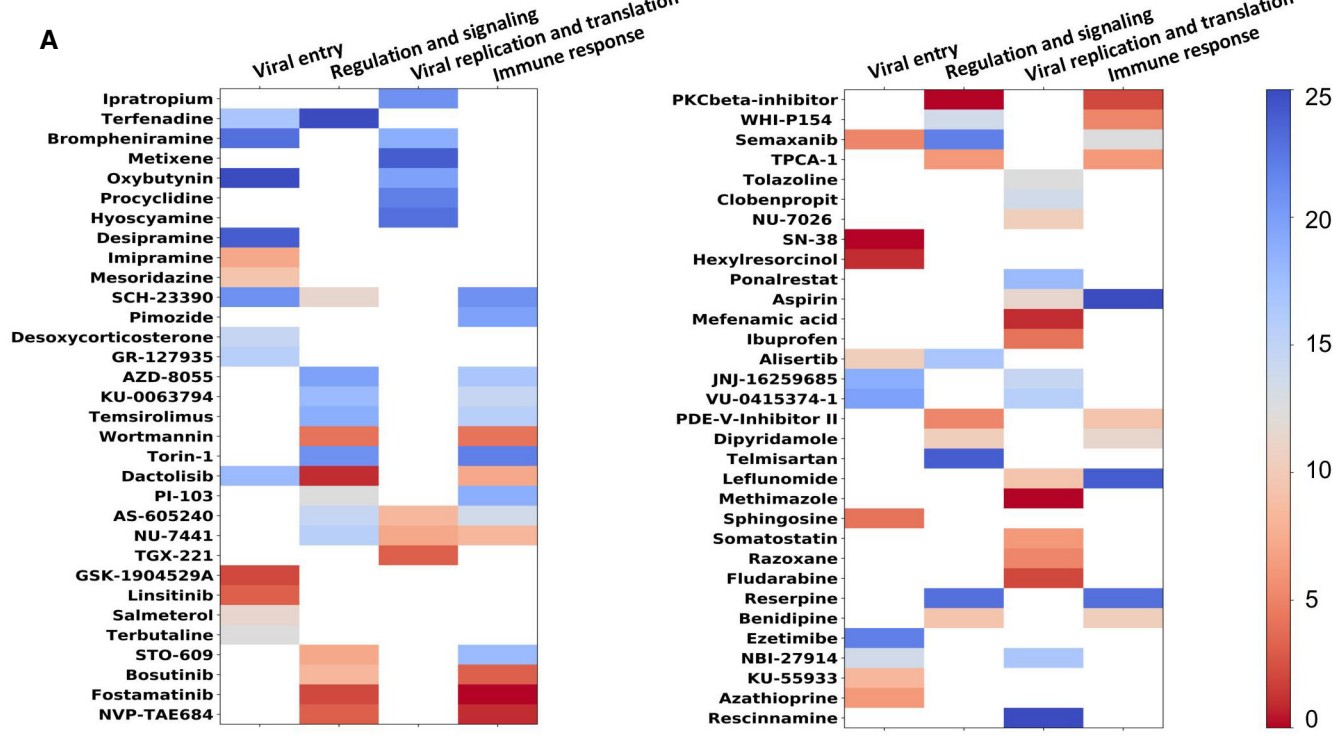

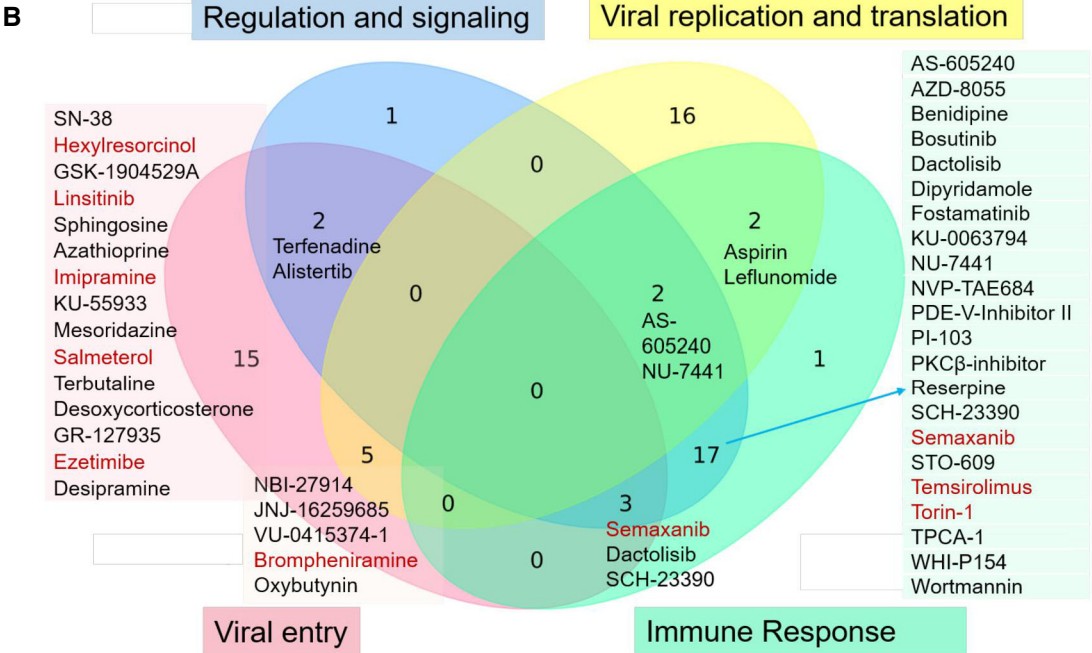

**Figure 3. Identification and classification of prioritized potentially antiviral compounds.**

A Prioritized compounds proposed to have potential antiviral activities and their involvement in different modules in the viral–host PPI network. 25 compounds/drugs were identified for each of the four modules, resulting in a total of 64 distinct repurposable drugs or investigational compounds, some participating in multiple modules. The entries in the heat map display the ranking, color-coded from red (highest) to blue (lowest). The ranking was based on the proximity of their targets to proteins belonging to the modules.

B Distribution of the same compounds/drugs in the four studied modules. Compounds belonging to selected intersections and to the viral entry module are listed. Those colored *red* have been experimentally tested. See the complete list in panel (A) and Appendix Table S5.

**Table 3. High-priority compounds with potential antiviral effects based on Dataset 1 (\*)**

**Prioritized compounds based on CMap scores and Network Proximity Ranks**

| Drug/compound[a] | Status | Disease module | Rank[b] | Description/MOAs | Ref |
|---|---|---|---|---|---|
| Brompheniramine* | FDA-approved | Viral entry | 23 | Histamine receptor antagonist | Gwaltney and Druce (1997) |
| Ipratropium | FDA-approved | Viral replication and translation | 21 | Acetylcholine receptor antagonist | Barnes (2004) |
| Imipramine* | FDA-approved | Viral entry | 8 | Norepinephrine and serotonin reuptake inhibitor, autophagy enhancer | Shchors et al (2015), Wichit et al (2017), Plenge et al (2020) |
| Temsirolimus* | FDA-approved | Immune response | 16 | mTOR inhibitor, autophagy enhancer | Di Benedetto et al (2010), Soliman et al (2013), Bergmann et al (2014), Kindrachuk et al (2015) |
|  |  | Regulation and signaling | 19 |  |  |
| Torin-1* | Investigational | Immune response | 22 | mTOR inhibitor, PI3K inhibitor, autophagy enhancer | Clippinger et al (2011), Bergmann et al (2014) |
|  |  | Regulation and signaling | 21 |  |  |
| AS-605240 | Investigational | Regulation and signaling | 15 | PI3K inhibitor, autophagy enhancer | Azzi et al (2012) |
|  |  | Viral replication and translation | 9 |  |  |
|  |  | Immune response | 14 |  |  |
| Linsitinib* | Investigational | Viral entry | 4 | IGF-1- and insulin receptor inhibitor, TBK1 activator through ARF1 | Mulvihill et al (2009), Sparrer et al (2017) |
| Salmeterol* | FDA-approved | Viral entry | 12 | β2 Adrenergic receptor agonist, autophagy enhancer | Medigeshi et al (2016) |
| Semaxanib* | Investigational | Viral entry | 6 | VEGFR inhibitor | O'Donnell et al (2005) |
|  |  | Regulation and signaling | 22 |  |  |
|  |  | Immune response | 13 |  |  |
| Hexylresorcinol* | FDA-approved | Viral entry | 2 | Local anesthetic | Wilson et al (1966) |
| Mefenamic acid | FDA-approved | Viral repl and translation | 2 | Cyclooxygenase inhibitor | Rothan et al (2016) |
| JNJ16259685 | Investigational | Viral entry | 19 | Glutamate receptor antagonist | Lavreysen et al (2004) |
| Ezetimibe* | FDA-approved | Viral entry | 22 | Niemann-Pick C1-like 1 protein antagonist, cholesterol inhibitor, autophagy enhancer | Osuna-Ramos et al (2018) |
|  |  | Regulation and signaling | 9 |  |  |

**Additional prioritized compounds (based on CMap scores and literature)**

| Drug/Compound | Status | Description/MOAs | Ref |
|---|---|---|---|
| QL-XII-47 | Investigational | Cytoplasmic tyrosine protein kinase BMX inhibitor | de Wispelaere et al (2020) |
| Rottlerin* | Investigational | MAPK and protein kinase inhibitor, autophagy enhancer | Lama et al (2019) |

[a]Those tested in experiments are indicated by asterisks in the first column.
[b]Rank refers to the proximity to the module in the third column, the lower the better.

compounds predicted to potentially act as viral entry blockers, i.e., imipramine, brompheniramine, linsitinib, semaxanib, and hexylresorcinol, in addition to salmeterol and ezetimibe from the above set (see Table 3). Our cell fusion assay, first described by Simmons et al (2004), detects host-cell-spike interactions on a shorter time scale than the viral infection assay and has been used by several groups to investigate the mechanisms of cell entry of SARS-CoV-1, such as endosomal and protease involvement including TMPRSS2 (Matsuyama et al, 2005; Matsuyama et al, 2010). More recently, the assay has also been used to investigate SARS-CoV-2-mediated cell entry (Ou et al, 2020). The assay is based on the principle that susceptible host cells ("acceptors") fuse with spike-expressing "donor" cells, forming large cell fusion constructs (syncytia), which can be quantified by fluorescence imaging.

We implemented this assay in a high-content, 384-well microplate format using HEK293T cells, which are not susceptible to viral infection unless transfected with ACE2 and TMPRSS2, and Calu-3 lung cancer cells, which possess the replete machinery for spike-mediated viral infection (Hoffmann et al, 2020a). HEK293T cells transfected with ACE2 and TMPRSS2 or native Calu-3 cells were incubated with donor cells co-expressing green fluorescent protein (GFP) and SARS-CoV-2 spike, and syncytia formation monitored by following GFP over time by fluorescence microscopy. After a 4-h incubation, syncytia were quantified by high-content analysis. Cell fusion was dependent on the presence of SARS-CoV-2 spike as donor cells expressing only GFP did not form syncytia. In a preliminary screen of seven computationally predicted compounds and two serine protease inhibitor positive controls (dec-RVKR-CMK and

**Table 4. Compounds proposed to help attenuate hyperinflammation based on Dataset 2.**

| Drug/Compound | Status | Description/MOAs | Ref |
|---|---|---|---|
| Compounds extracted from CMap and prioritized after QuartataWeb cluster analysis | | | |
| Midodrine | FDA-approved | Adrenergic receptor agonist | Josset et al (2010) |
| Olanzapine | | Dopamine receptor antagonist, autophagy enhancer | Altschuler and Kast (2020) |
| Trifluoperazine | | Dopamine receptor antagonist, autophagy dual-modulator | Ochiai et al (1991) |
| Fluphenazine | | Dopamine receptor antagonist, autophagy enhancer | Otreba et al (2020) |
| Azelastine | | Dopamine receptor antagonist, His receptor antagonist | Preprint: Konrat et al (2020) |
| Chlorphenamine | | Histamine receptor antagonist | Xu et al (2018) |
| Clarithromycin | | Bacterial 50S ribosomal subunit inhibitor autophagy inhibitor | Yamaya et al (2012), Pani et al (2020) |
| Saracatinib | Investigational | SRC inhibitor | Shin et al (2018) |
| **JAK3-Inhibitor-II** | | JAK inhibitor | Schwartz et al (2017) |
| **AZD-8055** | | mTOR inhibitor, autophagy enhancer | Jiang et al (2011) |
| CGP-60474 | | CDK inhibitor | He and Garmire (2020) |
| **Mepacrine/ Quinacrine** | | Cytokine production inhibitor, NFκB inhibitor | Dermawan et al (2014) |
| Hexamethylene | | Sodium/hydrogen antiport inhibitor | Wilson et al (2006) |
| Loperamide | FDA-approved | Opioid receptor agonist, autophagy enhancer | Shen et al (2019) |
| Nifedipine | | Calcium channel blocker, autophagy enhancer | Liu et al (2009), preprint: Straus et al (2020) |
| Liothyronine | | Thyroid hormone stimulant | Chen and An (2013) |
| Atorvastatin | | HMGCR inhibitor, autophagy enhancer | Episcopio et al (2019) |
| Triptolide | Investigational | RNA polymerase inhibitor, TNF-α inhibitor | preprint: Chaparala et al (2020) |
| Pirfenidone | FDA-approved | TGFβ receptor inhibitor, furin inhibitor, anti-fibrotic, autophagy enhancer | Ferrara et al (2020) |
| Oxaprozin | | Cyclooxygenase inhibitor, NSAID (non-steroidal anti-inflammatory drug) | Miller (1992) |
| Dexketoprofen | | Cyclooxygenase inhibitor, NSAID | Moore and Barden (2008) |
| Isoliquiritigenin | Investigational | Guanylate cyclase activator, autophagy enhancer | Traboulsi et al (2015) |
| PCA-4248 | | Platelet activating factor (PAF) receptor antagonist | Fernandez-Gallardo et al (1990) |
| Rucaparib | FDA-approved | PARP inhibitor, autophagy enhancer | Guo et al (2019) |
| Compounds extracted from CMap and prioritized by literature search | | | |
| Berbamine | Investigational | Calmodulin antagonist, autophagy inhibitor | preprint: Huang et al (2020) |
| Darinaparsin | | Apoptosis stimulant | Chowdhury et al (2020) |
| Taurodeoxycholic acid | | Bile acid | Li et al (2019a) |

The three drugs/compounds in boldface are also predicted as antiviral drugs based on Dataset 1, listed in Table 3.

nafamostat), pretreatment with dec-RVKR-CMK and nafamostat prevented syncytia formation (Appendix Figs S5 and S6), consistent with the involvement of those enzymes in spike-mediated viral entry (Ozden et al, 2008; Matsuyama et al, 2018; Hoffmann et al, 2020a). Notably, nafamostat, a potent wide spectrum serine protease inhibitor, has recently been found to inhibit the membrane fusion of SARS-CoV-2 at 15-fold higher efficiency than camostat mesylate (Hoffmann et al, 2020b). Dec-RVKR-CMK inhibits not only the enzymatic activity of furin but also those of cathepsin L, cathepsin B, trypsin, papain, and TMPRSS2 (Matsuyama et al, 2018). With the exception of semaxanib, all predicted compounds/drugs inhibited cell fusion to some extent, although some did so only at high concentrations (Appendix Figs S5 and S6).

Both agents that prevented viral infection in the experiments with Vero-E6 cells (salmeterol and ezetimibe), also had inhibitory activity in the cell fusion assay, although salmeterol was at least two orders of magnitude less potent in the cell fusion assay, and ezetimibe was inactive at the highest concentration tested in the viral infection assay, suggesting that their antiviral activity might not originate from an interference in viral entry, but other effects such as enhancement of autophagy, as discussed below. The most potent agent was the insulin-like growth factor 1 receptor (IGF1R) inhibitor, linsitinib. Inhibitor effects were qualitatively conserved in Calu-3 cells but generally more pronounced in transfected HEK293T cells (Appendix Figs S5 and S6). The one exception was the furin inhibitor dec-RVKR-CMK, which was similarly potent in both cell types but with a seemingly larger maximal magnitude of inhibition in Calu-3 cells, suggesting it inhibited other cellular pathways in addition to viral entry.

We then performed full dose–response curves in HEK293 cells with selected compounds (linsitinib, brompheniramine,

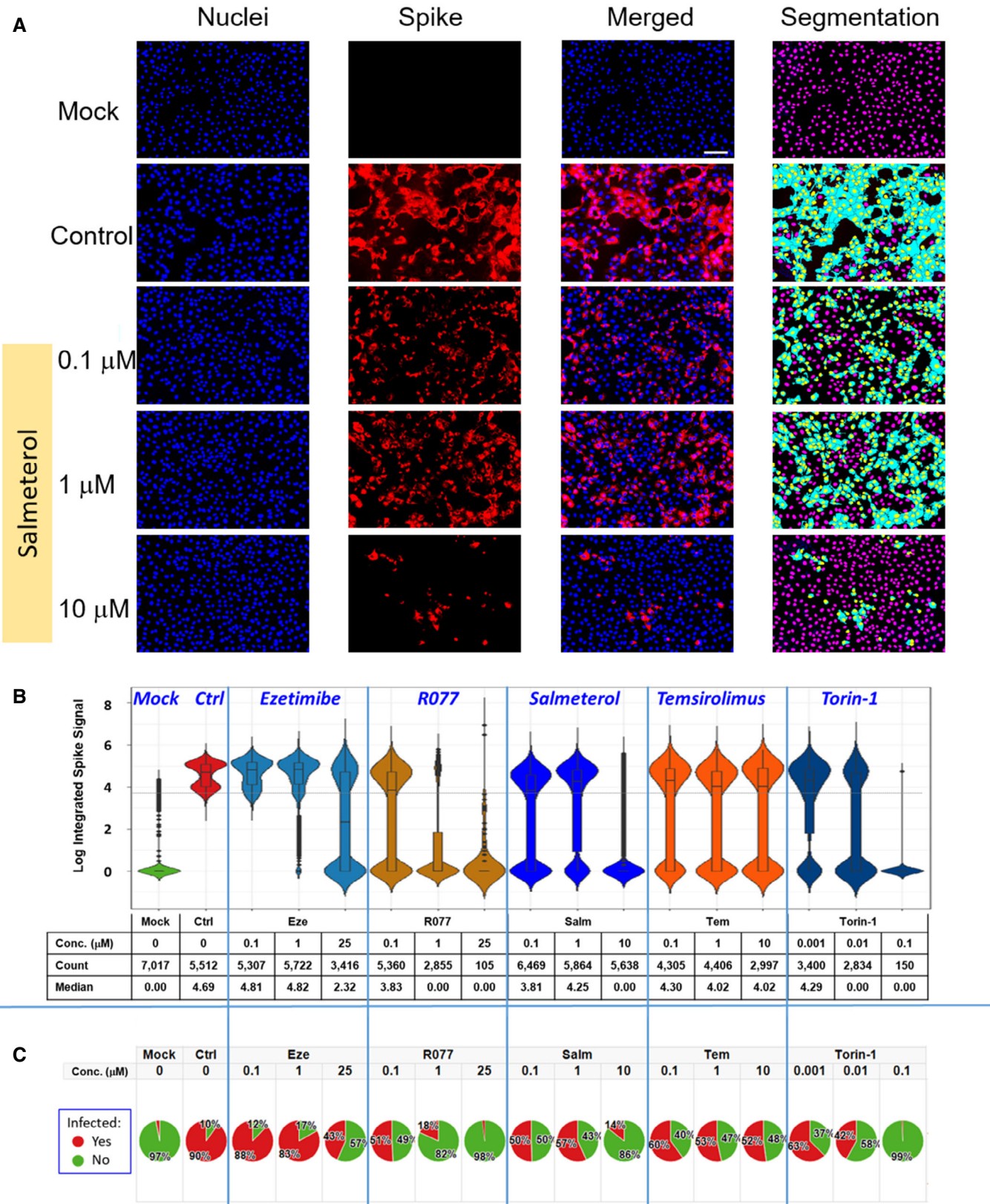

**Figure 4.**

**Figure 4. Suppression of SARS-CoV-2 infection by identified compounds.**

Vero-E6 cells were pretreated with compounds (salmeterol, rottlerin (RO77), temsirolimus, torin-1, or ezetimibe) for 1 h prior to SARS-CoV-2 inoculation. 48-h post-infection cells were fixed and fluorescently labeled for SARS-CoV-2 S protein.

A Representative fluorescence images of Mock, SARS-CoV-2 infected (Ctrl), and Salmeterol-treated wells analyzed with the Multiwavelength Cell Scoring application in MetaXpress. Grayscales of the images were adjusted to enable direct comparison of the relative levels of fluorescence among the treatments: Segmentation images show how cells were segmented and identified as spike positive. Purple, nuclei; cyan, spike. Scale bar, 100 μm.

B Violin plots of Vero-E6 cells labeled for Spike protein. The Multiwavelength Cell Scoring algorithm in MetaXpress was used to determine the integrated fluorescent signal in individual cells as a measure of the amount of Spike protein within each cell. The plots show the population distribution of the integrated signal for all of the treatments. The Boxes in the plot show the interquartile range (IQR) with the top and bottom edges marking the 75th and 25th percentiles, respectively. The horizontal line in the box is the median value, and the whiskers are defined to be 1.5 IQR. The ordinate is a log scale. The effect of the treatment is assessed quantitatively by changes in the median signal level, and qualitatively by observing changes in the modes. The dashed line is 3 standard deviations above the mean signal in the Mock samples and is used as a cutoff to quantify the number of cells that are positive or negative for the Spike signal. The statistics table below the plots shows the number of cells counted in each treatment group and the median of the population.

C Pie charts showing the effect of treatment on preventing infection of Vero-E6 cells. The number of cells above and below the cutoffs for being positive for Spike were counted and the percent cells in each category were determined. All analyses were done in Tibco Spotfire.

hexylresorcinol, and salmeterol), together with cytotoxicity assessments to test whether inhibition of syncytia formation could merely be a result of cell loss. Nafamostat, dec-RVKR-CMK, linsitinib, and to a lesser extent, brompheniramine, showed dose-responsive inhibition of syncytia formation that did not mirror cell loss (Fig 5). For example, linsitinib induced complete inhibition of cell fusion, whereas only partial cell loss was observed with a flattening of its dose–response curve. This quantitative and qualitative difference between the two dose–response curves suggests that the observed cell loss is likely to be an epiphenomenon, and not causing the inhibition of syncytia formation. In contrast, hexylresorcinol and salmeterol showed partial and full responses, respectively, on syncytia formation that were mirrored by cell loss (Fig 5). Further studies are required to determine in this assay with these particular drugs if cell loss (i) precedes inhibition of cell fusion thereby representing a nonspecific mechanism for preventing syncytia formation or (ii) is a specific result of inhibition of syncytia formation.

# Discussion

### Utility of the computational pipeline for identifying repurposable drugs

We presented here the results from a computation-driven approach for identifying repurposable drugs or new compounds that comply with the antiviral or anti-cytokine signatures derived from SARS-CoV-2-infected cells. The overall analysis was driven by the RNA-seq data from SARS-CoV-2-infected A549 cells and A549-ACE2 cells, as well as a SARS-CoV-2-host PPI network, toward gaining a system-level understanding of the key players in the host cell that are involved in SARS-CoV-2 infection and identifying potential modulators of these key players. Our extensive study led to 15 potentially antiviral and 23 potentially immune-modulatory compounds (Tables 3 and 4). The assays conducted to test ten of the proposed antiviral compounds pointed to several repurposable drugs or investigational compounds that could be pursued for lead development against SARS-CoV-2 infection. Among them, salmeterol exhibited particularly strong inhibitory activities in Vero-E6 cells infected by SARS-CoV-2 and linsitinib substantially reduced spike-protein-dependent syncytia formation (viral entry) in engineered HEK293T cells.

Recent studies point to the utility of computational systems pharmacology approaches for identifying repurposable drugs against SARS-CoV-2 (Beck et al, 2020; Gordon et al, 2020b; Riva et al, 2020; Singh et al, 2020; Zhou et al, 2020b,c,d). Of note is the work of Zhou et al (2020b) where repurposable drugs against SARS-CoV-2 were identified by evaluating the proximity of targets of known drugs to human proteins engaged in the human-CoV-host cell interactome. This type of network proximity analysis, originally introduced by Guney et al (2016), is also used here, but in a different context, mainly for prioritizing the candidate compounds/drugs that have been already identified from the DEG patterns of SARS-CoV-2 infected cells and corresponding CMap signatures. In contrast, Zhou et al (2020b) used gene set enrichment data (from MERS-CoV and SARS-CoV-infected cells) and CMap gene-drug signatures for validating their predicted drugs. Another important component unique to our analysis is the use of our interface QuartataWeb that allows for identifying drug-target associations, and for evaluating and classifying the pathways implicated in the disease modules deduced from the SARS-CoV-2-specific virus–host interactome (Gordon et al, 2020a,b) and assessing the mechanisms of action. QuartataWeb was further used to cluster the selected compounds based on their mechanisms of action and select representatives from each cluster to obtain a sufficiently diverse set for experimental testing. Thus, our study differs from that of Zhou et al (2020b) in the overall design of the computational protocol, the types of data used as input, as well as the output analyses for compound selection, prioritization, and validation, while both studies utilize state-of-the-art methods (network proximity analysis) and resources (e.g., CMap library) at different steps of the workflow.

Unlike influenza A and respiratory syncytial virus, the host immune defensive reactions of SARS-CoV-2 were significantly muted unless ACE2 was overexpressed (preprint: Blanco-Melo et al, 2020a). Cross-examination of the expression levels of the 17 anti-cytokine signature genes in A549 cells showed that most of these genes could not be clearly distinguished in those cells, i.e., their upregulation was specific to A549-ACE2 cells (compare panels (C) and (D) in Appendix Fig S7), whereas the 36 genes that define the antiviral signature exhibited a comparable expression pattern in A549-ACE2 cells (see panels (A) and (B) in Appendix Fig S7). These observations support the robustness of the antiviral signature on the one hand, and the utility of A549-ACE2 cells for detecting genes implicated in hyperinflammatory responses, on the other.

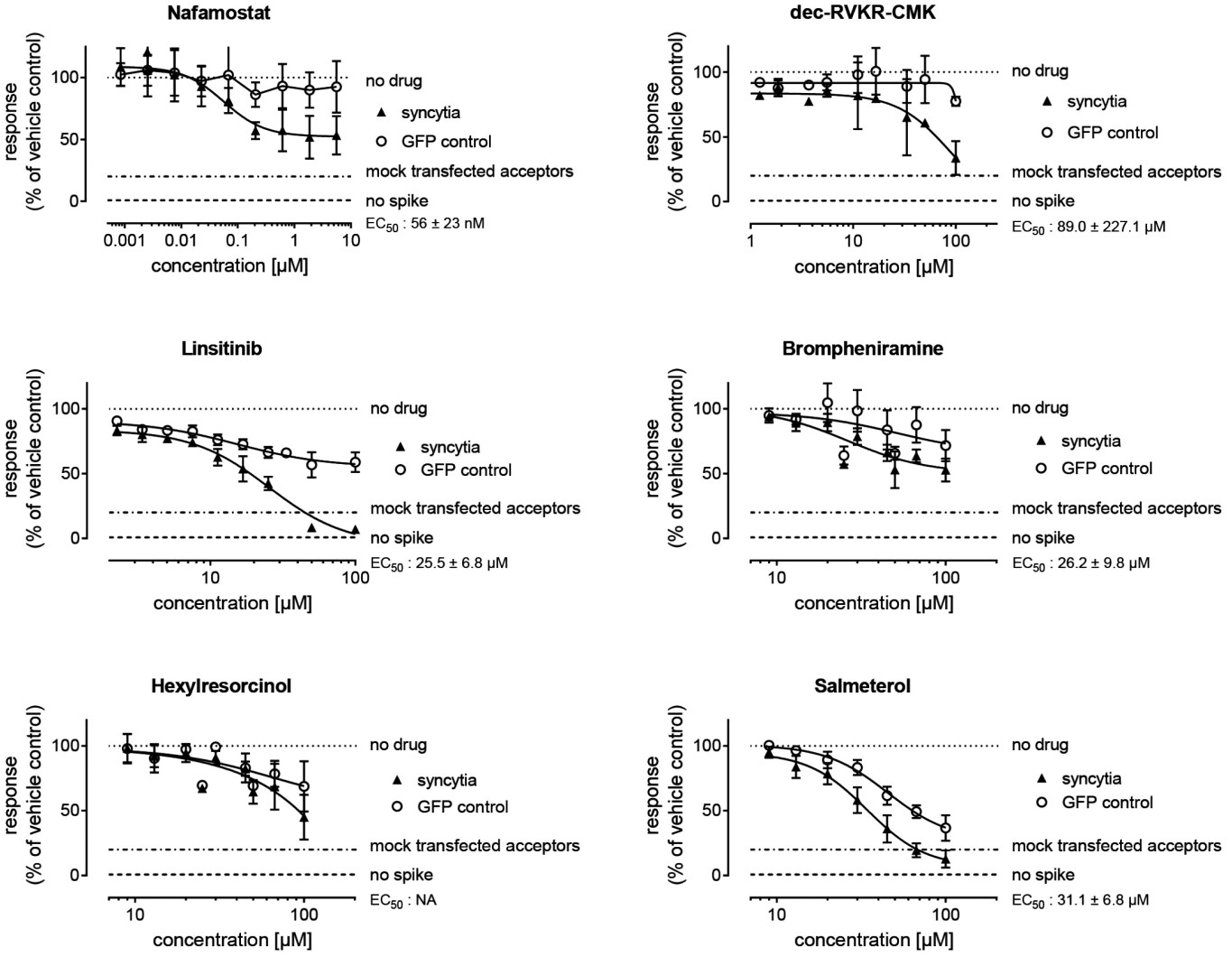

**Figure 5. Dose–response curves for selected compounds in the syncytia assay.**

HEK293 acceptor cells transfected with or without ACE2 and TMPRSS2 were seeded in 384-well plates, pretreated with 7-point gradients of test compounds for 1–2 h, and co-cultured for 4 h with HEK293 donor cells expressing SARS-CoV-2 spike and GFP, or donor cells expressing GFP only (no spike). Images of GFP-positive objects were acquired on a confocal high-content imager and analyzed for syncytia formation and integrated GFP area (total GFP) as a measure of cytotoxicity, using a CNT algorithm as described in the Materials and Methods. Data are the aggregate of 8 independent biological repeats; where errors are shown they represent SD from matching concentrations in at least three experiments.

**Potential mechanisms of action of drug candidates**

We performed two types of *in vitro* assays with ten predicted repurposable or investigational drug candidates most of which are proposed to be implicated in viral entry: linsitinib, imipramine, ezetimibe, hexylresorcinol, brompheniramine, salmeterol, semaxanib, rottlerin, temsirolimus, and torin-1. Viral entry is used here in a broad sense including (i) the fusion between viral and host cell membrane (involving ACE2 and $B^0AT1$ on the host cell membrane, and facilitated by host cell proteases such as TMPRSS2 and furin) and (ii) endosomal processes mediating the endocytosis of the virus and its release from the vesicles. The latter involves many signaling and regulatory proteins including those activated by the immune response, in addition to proteases

such as cathepsins, as schematically depicted in Fig 6A. The two experimental assays were chosen to complement each other: the viral infection assay recapitulates the entire virus infection process, whereas the syncytia assay addresses a specific, defined mechanism in viral entry, namely fusion of the virus with the host cell, which is mediated by interaction of viral spike protein with the host cell receptor (ACE2), and facilitated by host cell proteases.

Below we discuss the experimental results for the tested compounds in the light of their CMap scores, the similarities between their interaction patterns (as indicated by the clusters in Appendix Fig S2), the involvement of their targets in the host cell PPI network or disease modules (Fig 3) with reference to lung–tissue interactome (Fig 6B), and relevant findings from previous

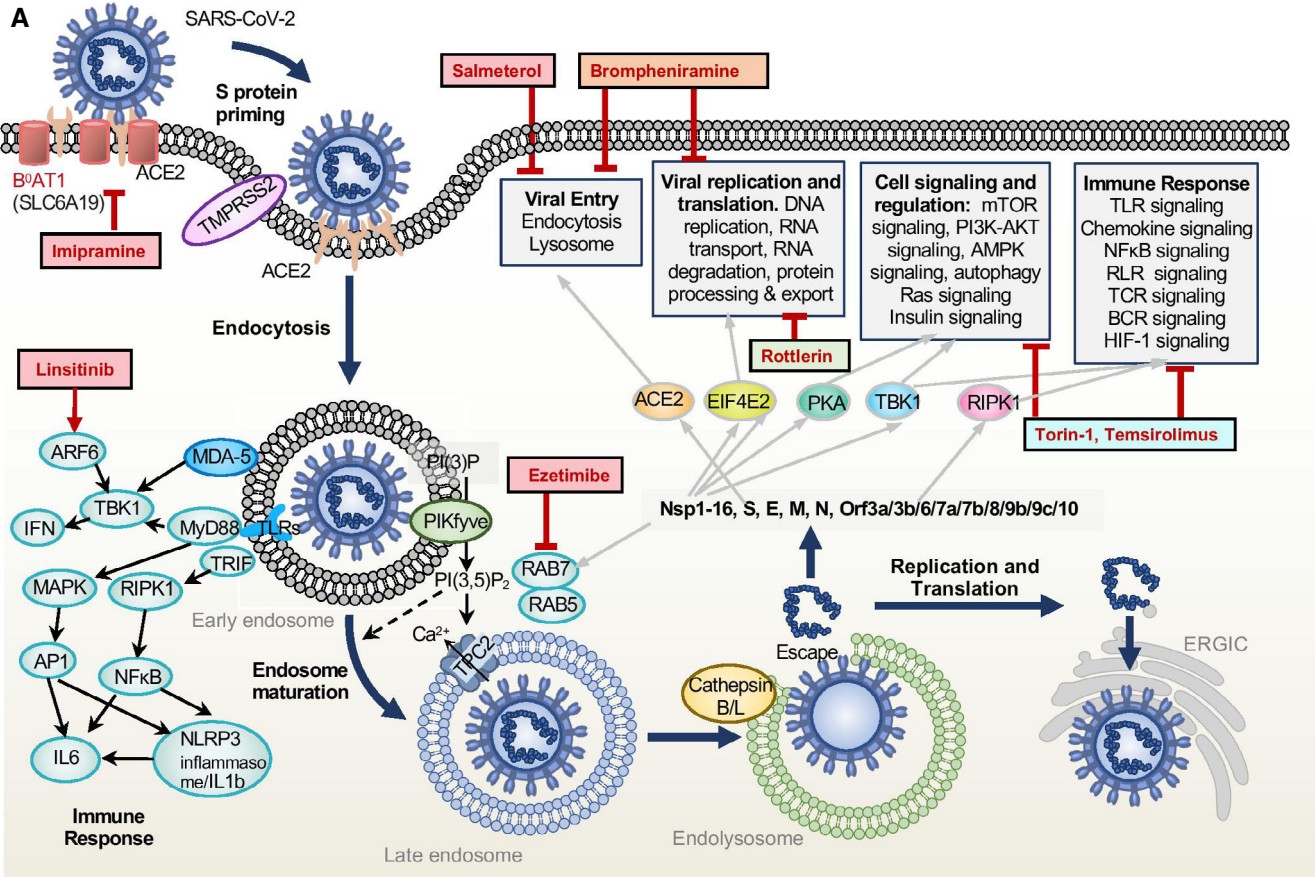

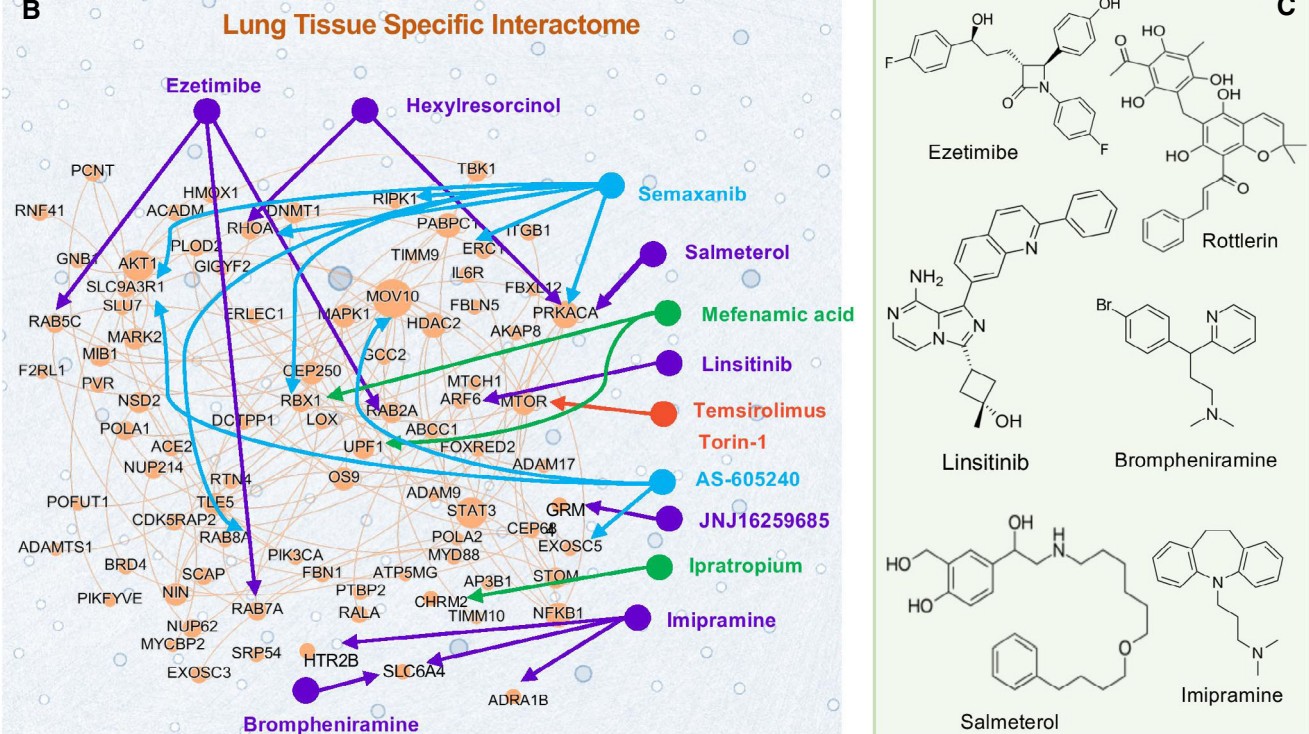

**Figure 6.**

**Figure 6.  Schematic representation of various stages of SARS-CoV-2 infection and selected compounds targeting various components of the viral–host interactome.**

A   Schematic description of viral entry, endosomal maturation, replication, translation, and accompanying cell signaling and regulation or immune responses, described in the main text. Mainly, SARS-CoV-2 spike binds the host receptor ACE2 (Hoffmann *et al*, 2020a) complexed with the amino acid transporter B0AT1 (Yan *et al*, 2020) Proteolytic cleavages (e.g., by TMPRSS2) are essential to viral entry, including spike priming and membrane fusion, or lysosomal escape after endocytosis. PIKfyve is the main enzyme synthesizing $PI(3,5)P_2$ in early endosome (de Lartigue *et al*, 2009), and $PI(3,5)P_2$ regulates early-to-late endosome events. TPC2 is a major downstream effector of $PI(3,5)P_2$ (Li *et al*, 2019b). Dominant pathways in four modules involved in SARS-CoV-2 infection are listed in the upper right boxes (see also Table 2). The diagram also shows selected drugs that have been identified and experimentally validated to inhibit or reduce SARS-2-CoV-2 infection (mainly viral entry) in highlighted in boxes (with red fonts).

B   Subnet of PPIs between host cell proteins implicated in SARS-CoV-2 infection and those targeted by selected compounds. The sandy brown nodes and edges represent the proteins and interactions in the SARS-CoV-2 host response network; and in the background (transparent light blue nodes and edges) is the lung tissue-specific protein interactome. The relative size of each protein node is consistent with its degree (number of connections) in the PPI network. Thirteen compounds we identified as candidate repurposable or investigational drugs for host-targeted antiviral therapy (based on Dataset 1) and their connections to targets in host response network (as reported in DrugBank or STITCH) are shown by color-coded labels and connectors. Magenta nodes represent the compounds that predominantly inhibit viral entry; light green and red represent those against viral translation, replication, and immune response; and *cyan* nodes represent multifunctional compounds.

C   Chemical structures of selected drugs displayed in panels (A) and (B); see all tested drugs in Appendix Fig S4.

work. We begin with compounds/drugs implicated in viral entry, as the focus of current tests (Fig 6C).

### Linsitinib

Linsitinib showed the highest inhibitory activity without overt cytotoxicity in the spike-induced syncytia formation assay that specifically measures viral entry. It is interesting to note that its proximity rank to the viral entry module (rank 4) was one of the highest among all tested compounds. Linsitinib is an IGF-1R and insulin receptor inhibitor (Mulvihill *et al*, 2009) currently under investigation for various types of cancer due to its ability to prevent tumor cell proliferation and induce tumor cell apoptosis (Fassnacht *et al*, 2015). Our analysis also indicated that it targets the insulin receptor, which interacts with ADP ribosylation factor 6 (ARF6), a binding partner of SARS-CoV-2 endonuclease nsp15 (Gordon *et al*, 2020b). As listed in Table 2, ARF is involved in multiple modules. Notably, the ubiquitination of the ARF domain of TRIM23 is essential for mediating virus-induced autophagy, an antiviral defense mechanism, via activation of TANK-binding kinase 1 (TBK1) (Sparrer *et al*, 2017). Therefore, we propose that its possible MOA is activation of TBK1 that promotes autophagy (see Fig 6A). We also note that while linsitinib was selected as a potential antiviral compound, it was also identified as an anti-inflammatory compound with a very high (−99.37) CMap score (Appendix Table S3B), in strong support of its selection as a high priority compound. In this context, the EC50 for linsitinib was 25 μM in the cell fusion assay that may not be disparate from the reported $C_{max}$ of 5–10 μM in patients (Macaulay *et al*, 2016). Since several IGF1/InsR inhibitors are available, this class of compounds is well suited for structure-activity studies. Such a study is particularly relevant, since CMap can implicitly account for structure-dependent non-canonical modes of antiviral activity that can differ among members of a particular drug class.

### Imipramine

Imipramine, an FDA-approved tricyclic antidepressant (Gillman, 2007), has been also reported to inhibit Chikungunya virus fusion (entry) (Wichit *et al*, 2017). It was distinguished by a high network proximity ranking (8[th]) in viral entry module (Table 3). Notably, imipramine is a high-affinity allosteric inhibitor of serotonin transporter (SLC6A4) (Plenge *et al*, 2020). Importantly, ACE2 is anchored into the host membrane through close association with the amino acid transporter, $B^0AT1$ (see Fig 6A). $B^0AT1$ is structurally homologous to serotonin transporter, sharing the LeuT fold typical of this family of sodium-coupled neurotransmitter transporters (Cheng & Bahar, 2019). Thus, imipramine is likely to also target $B^0AT1$, which may impair the ACE2-spike interaction, hence the observed inhibitory effect. In addition, imipramine has been reported to promote autophagy (Shchors *et al*, 2015), and this could be another (indirect) mechanism for alleviating SARS-CoV-2 infection.

### Brompheniramine

Brompheniramine is an FDA-approved drug known as a first-generation antihistamine drug, for treating common colds and allergic rhinitis (Simons *et al*, 1982). It shares a similar mode of action with imipramine, also targeting serotonin transporter. In our study, brompheniramine was indicated to be highly related to SARS-CoV-2 entry (ranked 23[rd] im the viral entry module). Both imipramine and brompheniramine inhibited syncytia formation, consistent with their hypothesized interaction with membrane-anchored ACE2.

### Salmeterol

Salmeterol had the highest CMap score for inducing the antiviral signature, and very high (network) proximity to the viral entry module. It is canonically used as a bronchial smooth muscle relaxant in asthma and COPD, as a long-acting β2-adrenergic receptor (β2-AR) agonist. COPD has been shown to be associated with increased expression of ACE2 (Leung *et al*, 2020), and a recent study on the effects of inhaled corticosteroids (ICS) on the bronchial epithelial cell expression of SARS-CoV-2-related genes in COPD patients demonstrated that a treatment with ICS in combination with salmeterol/fluticasone propionate decreased the expression of *ACE2* and *ADAM17* (preprint: Milne *et al*, 2020). We also note that β2-AR interacts with the PKA catalytic subunit α (Cα; encoded by *PRKACA*), which promotes autophagy-mediated degradation (Lizaso *et al*, 2013). Salmeterol has been reported to induce autophagy as a potential mechanism of inhibiting Dengue virus *in vitro* (Medigeshi *et al*, 2016). The observed inhibitory effect in Vero-E6 cells (Fig 4), which were not borne out by syncytia formation experiments with either HEK293T or Calu-3 cells, except at high concentration (Fig 5), is consistent with activities unrelated to viral entry, such as an innate immune response stimulation or autophagy enhancement.

### Ezetimibe

Ezetimibe, an FDA-approved lipid-lowering drug (Kosoglou *et al*, 2005), has a distinct MOA via the sterol transporter Niemann-Pick C1-Like 1(Nutescu & Shapiro, 2003). It targets sterol O-acyltransferase 1 (SOAT1) in the ER, which, in turn, interacts with the Ras proteins encoded by *RAB5C, RAB2A,* and *RAB7A*, implicated in early-to-late endosomal maturation. These proteins bind SARS-CoV-2 nsp7 (Gordon *et al*, 2020b). Loss of *RAB7A* (see Fig 6A and B) has been shown to reduce viral entry by altering endosomal trafficking and sequestering ACE2 inside cells (Daniloski *et al*, 2021). Finally, ezetimibe was also reported to interfere with the entry and replication of Dengue virus (Osuna-Ramos *et al*, 2018). In our hands, ezetimibe inhibited both viral infection and cell fusion. Its lower potency in the cell fusion assay is consistent with multiple mechanisms in addition to the dominant effect on viral entry, as described above.

### Hexylresorcinol

Hexylresorcinol ranked 2nd in the viral entry module. It is a FDA-approved over-the-counter product with anesthetic, antiseptic, and anthelmintic properties (Wilson *et al*, 1966) often used for upper respiratory irritations such as sore throat. It has sodium channel blocking effects and interacts with transglutaminase 2, a substrate of two SARS-CoV-2-related host proteins RhoA and PKA Cα. It also showed potential action against respiratory virus parainfluenza type 3 and cytomegalovirus (Shephard & Zybeshari, 2015). Yet, our *in vitro* cell fusion assay suggests that virus–host cell interactions may not be major contributors to its reported antiviral activities.

### Rottlerin

Rottlerin (R077), a natural polyphenolic compound, has been reported to inhibit influenza replication as an inhibitor of PKC (Hoffmann *et al*, 2008), and the translation of rabies virus circle by reducing intracellular ATP contents (Lama *et al*, 2019). It may have neuroprotective effects by its anti-oxidative and anti-inflammatory action in the central nervous system (Lee *et al*, 2020). Rottlerin inhibited viral infection but dose-limiting toxicity prevented a detailed analysis of viral entry vs. infection.

### Temsirolimus and torin-1

Temsirolimus and torin-1 are indicated to inhibit the protein kinase mTOR (Bergmann *et al*, 2014). The temsirolimus metabolite, sirolimus, as well as mTOR inhibitor rapamycin, are among the 128 approved drugs listed in the Excelra COVID-19 Drug Repurposing Database (Excelra, 2020). The PI3K-AKT-mTOR signaling pathway provides a cross-protective immunity against viral infection, especially against the influenza viruses (Lehrer, 2020), and has been recognized to regulate the translation and replication of coronaviruses (Zumla *et al*, 2016). mTOR inhibitors induce autophagy, which has been attributed to the inhibition of MERS-CoV (Gassen *et al*, 2019). Temsirolimus is currently FDA-approved for treating renal cell carcinoma (Miao *et al*, 2010). It has been reported to inhibit MERS-CoV infection (Kindrachuk *et al*, 2015). Torin-1 inhibits both mTORC1/2 complexes with $IC_{50}$ values between 2 and 10 nM and therefore was used at 1–10 and 100 nM levels and was toxic at 100 nM. Further studies will be required to determine the relative antiviral effects of these mTOR inhibitors in the context of their intrinsic dose-limiting toxicity.

### Semaxanib

Semaxanib a tyrosine kinase inhibitor, under development as a cancer therapeutic (O'Donnell *et al*, 2005), did not exhibit any inhibitory activity, despite its involvement in multiple modules.

### Compounds targeting immune response

Immunopathology of COVID-19 is longitudinally dynamic, individually diverse, more unique than other respiratory viral infections, and potentially detrimental when uncontrolled. It is featured with lack of interferon response, lymphopenia, and overwhelming inflammatory activation—especially in severe stage or patients with poor prognosis (Blanco-Melo *et al*, 2020b; Liu *et al*, 2020; Ong *et al*, 2020; Zhou *et al*, 2020a). Anti-cytokine therapeutics inhibiting IL-1 (NCT04324021, NCT0436281), IL-6 (NCT04320615, NCT04315298), TNF-α (Feldmann *et al*, 2020), or the broad-spectrum immune response by glucocorticoids (Lu *et al*, 2020) are currently investigated. Stemming from transcriptomic response following infection in A549-ACE2, we aimed for inducers that both elevate IFN signaling while suppressing cytokine pathways. The resulting compounds (Table 4), interestingly, included His receptor antagonists and TNFα inhibitors as expected, while also contained candidates such as PAF receptor antagonists, NFκB, SRC, JAK, and mTOR inhibitors, and neurological drugs blocking ion channels or neurotransmitter receptors. These results reveal the complexity of immune transcriptome modulation, involving heterogeneous states of multiple components and their coupled dynamics.

### Autophagy enhancement as a possible mechanism to exploit in combination therapies

The present analysis showed that certain autophagy-related vesicle pathways were downregulated, especially in the SARS-CoV-2-infected A549-ACE2 cells, which could be a potential escape mechanism from the immune system, as lysosomal digestion serves as an intrinsic antiviral program. These observations point to the opportunity of discovering drugs that exploit systems-level host response, i.e., stimulate autophagic response while suppressing hyperinflammatory responses. A recurrent pattern in several candidate compounds was indeed their involvement in autophagy enhancement. These include antidepressants as well as compounds repurposed to eliminate aggregates in the central nervous system, lung, or liver, such as trifluoperazine, fluphenazine (Table 4), and others (salmeterol and imipramine) that exhibited inhibitory activity in our experiments. Microglial autophagy has been recently pointed out to be essential for recovery from neuroinflammation (Berglund *et al*, 2020). In general, the role of autophagy in viral infection remains context-dependent, and both pathogen-destroying or viral-promoting effects have been reported (Maier & Britton, 2012), whereas inducing autophagy has markedly reduced MERS-CoV replication (Gassen *et al*, 2019). The effectiveness of selected autophagy enhancers observed here support their further investigation, at least in combination therapy, against COVID-19.

The compounds prioritized here targeted system-level modules, rather than individual targets. Beyond the urgent need for repurposing, these drugs can also be exploited as mechanistic probes to enhance our understanding of SARS-CoV-2 pathogenicity and drug resistance and provide a systems framework for developing combination therapies.

Comparison with earlier work showed that there are only nine compounds (apicidin, daunorubicin, entacapone, loratadine, metformin, mycophenolic acid, ribavirin, verapamil, and valproic acid) shared between our predictions and the recently reported 69 repurposable drugs (Gordon *et al*, 2020b). Given the little overlap with the drugs currently under clinical trials against SARS-CoV-2, the current findings may help complement the global COVID-19 drug discovery pipeline.

While we have adopted a systems-level approach, we also notice that we focused on the viral–host cell interactions that mediate viral entry and endosomal transitions, and on accompanying cell signaling and regulation events and immune response, in line with the assays conducted for probing viral entry. Events at the nucleus relevant to viral replication and translation play an equally important role, as evidenced by recent genome-wide CRISPR screens in Vero-E6 cells (Wei *et al*, 2021), which identified many proviral genes involved in chromatin regulation, histone modification, or epigenetic regulation. Compounds that target these specific pathways/processes, such as those involving the ubiquitous nuclear protein HMGB1 and the SWI/SNF chromatin remodeling complex (Wei *et al*, 2021) or the upregulation of cholesterol biosynthesis (Daniloski *et al*, 2021), are yet to be determined.

# Materials and Methods

### Evaluation of host-targeted antiviral and anti-hyperinflammatory signature from post-SARS-CoV-2 infection transcriptomics

The up- and downregulated gene list of A549 cells (human lung cancer) after 24 h of SARS-CoV-2 infection was obtained from GSE147507, and the corresponding DEGs were acquired from the DESeq2 result from the original publication with FDR adjusted *P*-value smaller than 0.05. This resulted in 100 upregulated and 20 downregulated genes listed in Appendix Table S1. Over-representation analysis was performed using gProfiler (Raudvere *et al*, 2019) with GO database (Carbon *et al*, 2019) for up- or downregulated genes, respectively, using Benjamini–Hochberg multiple test correction with a threshold of 0.05. Examination of the GO Biological Process (GO-BP) and GO cellular components (GO-CC) data for up- or downregulated genes resulted in 319 GO-BP and 13 GO-CC terms. The number of enriched upregulated terms was reduced by retaining those associated with no more than 300 genes, and not fewer than 10 overlapping genes, resulting in 16 GO terms (see column 6 in Appendix Table S2A). Downregulated terms were all kept. The enriched GO terms were organized and visualized with quickGO and classified as antiviral, proviral, or ambiguous. Those genes that defined the "antiviral signature" were obtained by merging the up- (innate immune response) or down- (intracellular vesicle) regulated antiviral genes and excluding proviral (viral genome replication) components. Genes classified as proviral or ambiguous were not included in the antiviral signature.

The resulting signature (composed of 36 genes) was used to screen for compounds/drugs in the L1000 database (Subramanian *et al*, 2017) which elicit a response that best matches the antiviral signature, reflected by their sufficiently high CMap connectivity scores, at https://clue.io/query. CMap scores range from −100 to 100, the two limits representing the least and the most similar compound-induced gene signatures, compared to our input antiviral signature. Compounds with top scores (in the suggested default range of 90–100) were selected for further analysis.

For the construction of anti-hyperinflammation signature, we focused on cytokine-related events (to be suppressed) by overlapping the GO cytokine response gene set (GO:0034097) with the upregulated genes (adjusted *P*-value < 0.05) from A549-ACE2-infected cells with high MOI of SARS-CoV-2 (GSE147507). We selected a final candidate set of 17 genes at the 0.05 upper quantile of $\log_2$ fold change (see Appendix Table S2B). This set of 17 genes was used as the upregulated gene input in CMap screening within the L1000 database, and the 275 compounds with lowest connectivity scores (varying from −90 to −100), showing strongest opposing effect, were selected.

### Identification of known compound-target interactions

The compound-target interaction search engine QuartataWeb (Li *et al*, 2020), which integrates STITCH (version 5) (Szklarczyk *et al*, 2016) and DrugBank (version 5.1.7) (Wishart *et al*, 2018), was used to identify targets for compounds obtained from CMap prediction. Specifically, all compound-target interactions recorded in DrugBank and the compound-target interactions with experimental confidence score no <0.4 in STITCH were integrated for further analysis. As a result, we retrieved 1,800 known interactions between 168 compounds and 746 targets, while no targets were identified for the remaining 95 compounds.

### Prioritizing the predicted compounds using their network proximity

The basic idea of network proximity (Guney *et al*, 2016) is to evaluate the significance of the network distance between a compound and a given disease module in the interactome. The methodology assumes that a compound is effective if it targets proteins within or in the immediate vicinity of a disease module. In our case, we extracted the human lung protein–protein interactome from the Biomedical Network Dataset Collection BioSNAP (Zitnik *et al*, 2018). We defined five viral-related modules, each containing a set (S) of pre-defined proteins derived from the host proteins implicated in SARS-CoV-2 infection (see the Results). For each compound, we have determined the set (T) of targets using QuartataWeb in the human lung PPI network. The proteins in sets S and T were connected via paths of zero or more intermediate protein nodes. Then we evaluated the distance between these targets and the pre-defined proteins from each viral-related module, in the human lung PPI network, as the average shortest distance path between the respective nodes *s* and *t* belonging to the sets *S* and *T*, as

$$d(S,\ T) = \frac{1}{\|T\|}\sum_{t \in T}\min_{s \in S} d(s,\ t).$$

Then, a reference distance distribution was constructed, corresponding to the expected distance between the disease module proteins and a randomly selected groups of proteins in the network, with the same size and degree of distribution as drug

targets in the network. This procedure was repeated 1,000 times, and the mean and standard deviation of the reference distance distribution were used to calculate a *z*-score by converting the observed distance to a normalized distance. Each compound was assigned a *z*-score with respect to each disease module, a lower *z*-score meaning that its targets were closer to the disease module, or the compound would be more effective. The *z*-scores were evaluated using the toolbox package developed by Guney *et al* (2016). Note that the network proximity provides a relative measure, the absolute value of which depends on the disease and application. In the current application to four disease modules, we refrained from selecting a uniform cutoff for the *z*-score. Instead, we selected the top 25 compounds from each module to include a set of compounds with diverse MOAs.

## Compound clustering by means of interaction-pattern-based similarities

We clustered top-ranking compounds by evaluating the similarities between the interaction patterns of these compounds vis-à-vis their known targets compiled in DrugBank and STITCH. Specifically, we assigned each compound $i$ a vector $\boldsymbol{u_i}$ the elements of which were the confidence score for the compound–target interaction (0 if there is no known interaction). Then, we evaluated the interaction-pattern-based similarities between compound $\boldsymbol{i}$ and $\boldsymbol{j}$ by calculating cosine distance between vector $\boldsymbol{u_i}$ and vector $\boldsymbol{u_j}$ using the similarity metric $s = 1 - (\boldsymbol{u_i} \cdot \boldsymbol{u_j}) / (|\boldsymbol{u_i}| |\boldsymbol{u_j}|)$.

## *In vitro* viral inhibition assays

SARS-CoV-2 viral assays were performed in UCLA BSL3 high containment facility. Vero-E6 [VERO C1008 (ATCC# CRL-1586™)] cells were obtained from ATCC and cultured at 37°C with 5% $CO_2$ in EMEM growth media with 10% fetal bovine serum and 100 units/ml penicillin. SARS-CoV-2 Isolate USA-WA1/2020 was obtained from BEI Resources of National Institute of Allergy and Infectious Diseases (NIAID). Temsirolimus (CAS 162635-04-3), Ezetimibe (CAS 163222-33-1), Salmeterol (CAS 89365-50-4), and Torin-1 (CAS 1222998-36-8) were purchased from Selleckchem. Rottlerin (CAS 82-08-6) was purchased from TOCRIS. Vero-E6 cells were plated in 96-well plates ($5 \times 10^3$ cells/well) and pretreated with compounds (in triplicate, at indicated concentrations) for 1 h prior to addition of SARS-CoV-2 (MOI 0.1). After 48-h post-infection (hpi) the cells were fixed with methanol for 30–60 min in −20°C. Cells were washed three times with PBS and permeabilized using blocking buffer (0.3% Triton X-100, 2% BSA, 5% Goat Serum, 5% Donkey Serum in 1 × PBS) for 1 h at room temperature.

Subsequently, cells were incubated with anti-SARS-CoV-2 Spike antibody (Sino Biological, 40150-R007, 1:200) at 4°C overnight. Cells were then washed three times with PBS and incubated with Goat anti-mouse IgG Secondary Antibody, Alexa Fluor 555 (Fisher Scientific PIA32790, 1:1,000) for 1 h at room temperature. Nuclei were stained with DAPI (4′,6-Diamidino-2-Phenylindole, Dihydrochloride; Life Technologies) at a dilution of 1:5,000 in PBS for 10 min. Cells were analyzed by fluorescence microscopy. Five images per well were quantified for each condition. The Multiwavelength Cell Scoring module in MetaXpress (Molecular Devices, Sunnyvale, CA) was used to measure the total integrated

fluorescence spike signal in each cell. Histograms of the log of the integrated intensities were plotted in Spotfire (Tibco, Palo Alto, CA). A cutoff value of three standard deviations of the total integrated signal from the mock samples was established, above which cells were considered to have a positive spike signal, and thus be infected. The number of infected cells was divided by the total number of cells in each treatment group to determine the percent of infected cells after treatment.

## Cell fusion (syncytia) assay

### *Cell culture*
HEK293T cells (ATCC CRL-3216) were maintained at 37°C in a humidified incubator with a 5% $CO_2$ atmosphere. Cells were cultured in Dulbecco's modified Eagle medium (DMEM, Gibco 11965092) supplemented with 10% fetal bovine serum (FBS, Corning 35010CV), 1% penicillin–streptomycin (Cytiva HyClone SV30010), and 1% L-glutamine (Cytiva HyClone SH3003401). A cell bank of defined passage was established, and cells were propagated for no more than 15 passages in culture. A cell bank of Calu-3 cells (ATCC HTB-55) from cells maintained in DMEM as recommended by ATCC was established at early passage. Because Calu-3 cells grew very slowly in DMEM, for experiments cells were switched to Roswell Park Memorial Institute (RPMI) 1640 (Cytiva HyClone SH30027.01), which provided much better growth conditions. All cell lines were routinely tested for mycoplasma infection and passaged no more than 10 times from ATCC authenticated stocks.

### *Reagents*
Expression plasmids for human ACE2, TMPRSS2, and HA-tagged SARS-CoV-2 spike were a gift from Stefan Pöhlmann (Hoffmann *et al*, 2020a). Dec-RVKR-CMK (furin inhibitor-1) was from EMD Millipore (344930). Imipramine hydrochloride, Salmeterol, and Brompheniramine were from AK Scientific (J10511, K-590, and M-1266, respectively). Hexylresorcinol, Semaxanib (SU-5416), Ezetimibe, and Linsitinib (OSI-906) were from TargetMol (T0314, T2064, T1593, and T6017, respectively).

### *Transfection of cells for syncytia assay*
On the day of experiments, acceptor cells were transfected with mammalian expression plasmids for ACE2 and TMPRSS2 using FuGene6 (Roche) at a 1:3 DNA-to-reagent ratio with 22 ng DNA per well (30 μl) of a 384-well plate. 4,000 cells were plated in collagen-coated microplates (Greiner 781956) and centrifuged at 500 *g* for 1 min. Donor cells were transfected under the same conditions with expression plasmids for eGFP or eGFP plus SARS-CoV-2 spike protein and plated in T-25 flasks (3 ml). Both donor and acceptor cells were incubated for 3 days at 37°C. Calu-3 cells were left untransfected and seeding density was 8,000 cells/well in RPMI.

### *Cell treatment for High Content Screening*
On the day of co-culture, acceptor cells were pretreated for 1–2 h with vehicle or test agents; compounds were dissolved in DMSO and diluted into complete DMEM to a 3× concentration of the highest desired concentration in the assay. The resulting solutions were serially diluted on a 96-well plate into DMEM containing 3% DMSO. Fifteen microliter of the resulting gradients were transferred to cells using a Biomek 2000 liquid handler (Beckman Coulter) in duplicate

to yield quadruplicate measurements for each concentration of test agents. The final concentration of DMSO in the assay was 1%. Each plate contained 80 wells of vehicle controls, 16 wells of mock-transfected acceptor cells, and 16 wells of ACE2/TMPRSS2 transfected acceptor cells incubated with GFP-only expressing donor cells (no spike).

### Syncytia assay co-culture, imaging, and analysis

Donor cells were dislodged from their flasks with non-enzymatic cell dissociation buffer (Thermo Fisher 13151014) after two gentle washes with PBS. GFP-positive cells were counted in a hemocytometer. 2,000 GFP-positive cells in 15 µl DMEM were added to acceptor cells, plates centrifuged at 500 *g* for 1 min, and syncytia formation monitored. After 4 h cells were imaged live in the GFP channel (Ex485/Em525 nm) on a Molecular Devices ImageXpress Ultra or a Perkin Elmer OPERA Phenix Plus High Content Screening (HCS) reader using a 20X objective. Four fields were acquired per well. Images were uploaded to Definiens Developer (Ver 6, Definiens AG, Germany) and analyzed by a custom Cognition Network Technology (CNT) ruleset that separated individual cells, cell aggregates, and syncytia based on size, intensity, and texture of GFP expressing objects. The final parameters used for plotting were the percentage of GFP-positive area covered by syncytia relative to the total area covered by GFP-positive objects, and the total GFP-positive area as a surrogate for cell number. Data were averaged from the four imaging fields and normalized to vehicle-treated controls. Data from multiple independent experiments were pooled and analyzed by one-way ANOVA followed by Dunnett's multiple comparisons test. Dose–response data were fitted to a four-parameter logistic equation in GraphPad Prism (Ver. 7).

## Data availability

The data and codes generated during the study are available at: https://github.com/Hannah-Qingya/Covid19_systems-level_analysis. We also used our QuartataWeb server that is online accessible at http://quartata.csb.pitt.edu/.

**Expanded View** for this article is available online.

### Acknowledgements

S.Y.C. is supported by the American Heart Association Established Investigator Award 18EIA33900027 (S.Y.C.) and the American Lung Association Award ETRA734979 (S.Y.C.). D.L.T. is supported by RO1DK117881, UG3 DK119973, and a COVID-19 supplement to UG3 DK119973. I.B. is supported by NIH grants P41 GM103712 and PO1 DK096990. This project used shared instrumentation that was acquired with NIH grant S10 OD028450.

### Author contributions

Conceptualization, FC, QS, FP, BL, and IB; Methodology, FC, QS, FP, AV, RAP, GGJ, ACG, MHC, MS, BL, SYC, VA, AMS, MA, and IB; Validation, FC, QS, FP, AV, RAP, MS, AMS; Formal Analysis, FC, QS, FP, AV, RAP, GGJ, ACG, MS, BL, SYC, VA, AMS and IB; Investigation, FC, QS, FP, AV, RAP, GGJ, ACG, MHC, MS, SYC, VA, AMS, MA, and IB; Resources, AV, SYC, VA, AMS, DLT, MA, and IB; Data curation, FC, QS, FP, AV, RAP, GGJ, ACG, MHC, MS, SYC, VA, AMS, MA, and IB; Visualization, FC, QS, FP, AV, RAP, MHC, MS, BL, AMS, and IB; Writing, FC, QS, FP, AV, AMS, MA, and IB; Supervision, AV, AMS, MA, and IB.

### Conflict of interests

S.Y.C. has served as a consultant for United Therapeutics; and S.Y.C. has held research grants from Actelion and Pfizer. S.Y.C. is a founder, director, and officer in Synhale Therapeutics. The other authors declare no conflict of interests regarding the publication of this paper.

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
