## [Review Process File · Molecular Systems Biology]

A systems-level study reveals host-targeted repurposable drugs against SARS-CoV-2 infection

Fangyuan Chen, Qingya Shi, Fen Pei, Andreas vogt, Rebecca Porritt, Gustavo Garcia Jr, Angela Gomez, Mary Cheng, Mark Schurdak, Bing Liu, Stephen Chan, Vaithilingaraja Arumugaswami, Andrew Stern, D. Lansing Taylor, Moshe Ardit, and Ivet Bahar

DOI: [10.15252/msb.202110239](https://doi.org/10.15252/msb.202110239)

Corresponding author(s): Ivet Bahar (bahar@pitt.edu)

Review Timeline:

Submission Date:	21st Jan 21
Editorial Decision:	19th Feb 21
Revision Received:	27th Apr 21
Editorial Decision:	1st Jun 21
Revision Received:	10th Jun 21
Accepted:	11th Jun 21

Editor: Jingyi Hou

Transaction Report:

Thank you for submitting your work to Molecular Systems Biology. We have now heard back from the three reviewers who agreed to evaluate your manuscript. As you will see from the reports below, the reviewers acknowledge the potential interest of the study. However, they raise a series of concerns, which we would ask you to address in a major revision.

Since the reviewers' recommendations are rather clear, there is no need to reiterate all the points listed below. In particular, during our pre-decision cross-commenting process (in which the reviewers are given a chance to make additional comments, including on each other's reports), Reviewer #1 added, "I have to enhance that there is lack of novelty of methodologies proposed in this study. Host-target therapies idea, network proximity analysis, and CMAP gene signature analysis are published by several recent COVID-19 drug repurposing studies, such as <https://www.nature.com/articles/s41421-020-0153-3> and <https://journals.plos.org/plosbiology/article?id=10.1371/journal.pbio.3000970>. Yet, they didn't provide credits for these previous studies and call these approaches a new systems pharmacology approach in this manuscript. Although they provide experimental validation, the drug doses they tested are very high, reducing enthusiasm of potential clinical application of tested drugs." Reviewer #3 said "I wasn't aware of the two studies mentioned by Reviewer 1 in his/her comment. In the light of these work, I would certainly recommend to the authors to give credit to these two previous studies and to extensively detail in the response to reviewers, as well as in the main text, differences and commonalities with respect to their approach."

In light of the comments of the reviewers, some of the key issues that would need to be addressed are the following:

- The study should be placed in the context of previous literature, and effort should be made to clearly highlight the novelty of this study.
- Reviewer #1's concerns with regard to the experimental validations need to be carefully addressed to better support the conclusions of the study.

All other issues raised by the reviewers need to be satisfactorily addressed as well. As you may already know, our editorial policy allows in principle a single round of major revision and it is therefore essential to provide responses to the reviewers' comments that are as complete as possible.

REFEREE REPORTS

Reviewer #1:

The authors adopted a computational framework to prioritize possible treatments for COVID-19 and experimentally validated compound/drug candidates using Vero-E6 cell models. The computational framework, including network proximity analysis and transcriptome-based gene enrichment analysis, have been reported by two previous studies for COVID-19 drug repurposing, including Zhou et al., Cell Discovery 2020 and Zhou et al., PLOS Biology 2020. The authors have to highlight new contributions of this study compared to previous studies from methodology perspectives, not call this published approach a new quantitative systems pharmacology (QSP) approach. Although the authors performed experimental validations for some predictions, current experimental assays were very preliminary and more quantitative indices to quantify inhibitory of compounds (such as IC50 and EC50 values) should be provided. Another major concern is that dose/concentration tested in Figure 6 is really high, which is out of physiological concentration.

1. As shown in Figure 1, the proposed QSP approach is very similar with two previous publications, Zhou et al., Cell Discovery 2020 and Zhou et al., PLOS Biology 2020. The authors should highlight new contributions of this study compared to previous Zhou's methodologies.
2. Figure 1G was copied from the previous network proximity paper published by Zhou et al., Cell Discovery 2020.
3. It is not clear how the authors selected candidate drugs/compounds for experimental validation. For example, which specific z-score cutoff from network proximity analysis are used.
4. For network proximity analysis, it is unclear how the authors calculate network proximity using which human protein-protein interaction networks. The reviewer cannot find any information about human protein-protein interaction networks.
5. More details for methodology and rigor should be provide to improve reproducibility.
6. As shown in Figure 5, the authors only shown suppression activities for predicted compounds. The authors should provide standard quantitative indices to quantify inhibitory of compounds, such as IC50 and EC50 values.
7. As shown in Figure 6, most compounds didn't show any dose-dependent responses in syncytia formation assay. In addition, dose/concentration tested in Figure 6 is really high, which are out of physiological concentration.
8. For investigational molecules, the authors should discuss toxicity and pharmacokinetic profiles.

Reviewer #2:

The authors present a nice systems pharmacology method to identify drugs, targeting the host cell, with anti-SARS-CoV-2 activity and also drugs to inhibit the hyperinflammatory state in the late phase of COVID19 disease. Several of the identified drugs are experimentally validated. Identification of repurposable, host targeting drugs is an important field of the current SARS-CoV-2 related research, and the proposed pipeline theoretically can also be used in other viral infection diseases.

Detailed comments and questions:

Major comments

- The authors selected 36 genes from the 120 differentially expressed (DE) genes for further analysis - "Therefore, after careful evaluation, we selected 36 genes to be upregulated (Figure 2C), comprised of 26 upregulated genes associated with viral defense, and 10 downregulated genes associated with endocytic or vesicular processes." Could the authors more detailed the steps of "careful evaluation"?
- The authors write, that "The A549-ACE2 cells (Dataset 2) repeatedly exhibited a more pronounced cytokine upregulation, along with IFN response insufficiency, compared to A549 cells." The more strong response of A549-ACE2 cell line could be the consequence of the presence of ACE2 (viral receptor) on these cells, which should be discussed. Also, it is not entirely clear for me, how the authors concluded, that the gene signature of A549 cells can be used for antiviral, while the signature of A549-ACE2 cells can be used for anti-inflammatory signal generation. It would be also interesting to see how the 36 genes of antiviral signature (from A549 cells) and the 17 genes of anti-inflammatory signature (from A549-ACE2) change in the other cell lines (A549-ACE2 and A549 respectively).

Minor comments

- The authors use not only the classical "signature-reversal" hypothesis in their work, but considered that drugs activating the host cells antiviral mechanisms can be also effective antiviral drugs. This is a biologically motivated hypothesis, which shows to be correct later in the manuscript. Several other studies investigated drug repurposing against SARS-CoV-2 using the "classical signature-reversal" hypothesis, without much experimental validation, so I found the method of the authors novel and elegant.
- For the antiviral signature, the authors identified 100 up and 20 downregulated genes in infected A549 cells. The authors state that the upregulated genes are related to IFN response, however they state that "induction of chemokine, cytokine, and interferon types I and II were more 'muted' in SARS-CoV-2-infected A549 cells compared to those of other respiratory viruses such as influenza A and respiratory syncytial virus (Blanco-Melo et al., 2020a)". Based on which analysis did they reach this conclusion (i.e. the muted chemokine / cytokine / interferon response)?
- In Figure 5B the authors show the results of the viral infection assay. I think a violin or swarm plot would be more suitable for displaying the results than the current boxplot + distribution plot.
- I generally felt the antiviral part of the study very convincing, both from the side of systems pharmacology method and the experimental validation. On the other side the anti-hyperinflammatory part is weaker in my opinion. At first, the authors also used the virus-host protein interactions and a lung specific PPI network (if I understand correctly) to prioritise this list. I think the COVID19 related hyperinflammatory state is more related to different immunocytes than the originally infected lung cells. Despite these (theoretical) considerations, the identified drugs looks promising eg. glucocorticoid agonists and TNFa inhibitors. However I was not able to find glucocorticoid agonists and TNFa inhibitors in the corresponding Table4. Also, some (at least literature-based) validation of these drugs would be also suitable.

Reviewer #3:

The authors present results from an original system pharmacology method aiming at identify small molecules and drug repositioning opportunities for treating SARS-CoV-2 infected patients.

The presented method is based on the nowadays famous cMap database and relative signature matching approach successfully used in previous works based on a 'signature reversion principle'. In most of these works a transcriptional signature characterising a disease phenotype to be rescued is first identified then used to find drugs exerting an 'inverse' effect on cellular transcriptional program, i.e. up-regulating genes that are down-regulating in the disease signature and vice-versa. The hypothesis is that reverting the disease signature might revert (thus rescue) the disease phenotype itself. This approach has been proven effective in a number of works and formally demonstrated at least for metabolic disorders.

Here the authors go one step further, designing a sort of 'rationale connectivity mapping' approach where the transcriptional signature summarising the phenotype to be rescued or (as in this case) just modulated is generated from two condition specific sets of differentially expressed genes (from publicly available RNAseq data) are assembled, combined and then refined to identify a component that should be reverted (the anti-viral component) and another one that should be enhanced (to reduce inflammatory host-response). This is accomplished via a knowledge based (or semi-supervised) approach based on the observation and selection of relevant enriched GO categories).

By using these signatures (assembled from publicly available data) against the cMap, the authors are able to identify large sets of hit compounds that are further prioritised using a network proximity based approach (using a recently published SARS-CoV-2 specific network). Shortlisted hits are finally successfully experimentally validated.

Briefly this is a nice piece of work that extends an established computational paradigm in an innovative and original way and whose results are robustly validated and might have a great impact.

Few major points should be addressed before further considering this manuscript for publication in Molecular Systems Biology

Major points:

- whereas the idea of a 'rationale' or semi-supervised connectivity mapping is valid and justified, the GO term guided selection of the genes to be included in the final query to the cMap appears to overwhelm the transcriptional data, with (for example) a mixture of up/down regulated genes included in the anti-viral component of the final signature. What would happen if starting just from intersecting all relevant GO categories to compose the set of genes to be up- down-regulated by the cMap compounds? The author should convincingly show that a pure knowledge based approach just based on GO categories or signatures from the MsigDB database would lead to different results and set of compounds outputted when querying the cMap and that there is indeed value in combining such approach with the initial enrichment analysis of the genes differentially expressed upon viral infection.
- components 3 and 4 of figure 1A seem to refer only to the genes to be upregulated. Why there isn't an equivalent couple of visuals for the 17 genes to be downregulated?
- it is not immediately clear if the two components of the signature were used as a single query or into two separate instances. This should be explicitly mentioned. The authors report two sets of compounds outputted by their query: a set of potentially anti-viral compounds and a set of potentially anti-cytokine compounds, hinting that there were two queries, but they also wrote 'our query/input signature'. This is very confusing.
- The benchmark output against Excelra, QuartataWeb etc. is important and should be summarised somewhere in a figure panel.
- There is a lack of statistical assessment of the target proximity to the disease relevant modules

identified in the SARS-CoV-2 specific interaction network. What is the expected distance of randomly selected nodes to any of the modules? The author should use a less arbitrary threshold possibly based on this assessment to further prioritise compounds instead of just picking the top ranking 25 compounds etc.

- The discussions section is long and quite boring. I do not believe that it is necessary to list all those details for the identified compounds. The authors should limit to discussing their approach and findings and potential implications.

Minor points:

- in the introduction the authors claim that "Many compounds under clinical trials against SARS-CoV-2 are repurposable drugs". Given that the compounds the authors are referring to are still under clinical investigation, this sentence should be amended as "Many compounds under clinical trials against SARS-CoV-2 are potentially repurposable drugs"

- 'Quantitative systems pharmacology' is used 4 times through all the manuscript. Does this really require introducing an acronym? which is confused and also used in a section title (which should be generally avoided).

- A cMap based drug signature refinement approach aiming at improving drug repositioning predictions and moving in the same alley of the approach presented in this manuscript has been introduced recently and could be cited (PMID: 26452147).

Response to reviewers

We thank the reviewers for their thoughtful and thorough reading and evaluation of our manuscript. In particular, Reviewer 1 commented on the preliminary nature of some of our experimental assays, which we have addressed in the revised version. Specifically, we are now providing full dose-response curves for the syncytia assay, with canonical curve fitting parameters, on the most promising agents in the preliminary experiments presented in the original manuscript, and we have incorporated a cytotoxicity measurement to rule out artifacts. Below are our point-by-point responses to reviewers' comments. The reviewers' comments are italicized, our responses are in plain font, and excerpts from the manuscript are in blue, different font, and indented.

Reviewer #1:

The authors adopted a computational framework to prioritize possible treatments for COVID-19 and experimentally validated compound/drug candidates using Vero-E6 cell models. The computational framework, including network proximity analysis and transcriptome-based gene enrichment analysis, have been reported by two previous studies for COVID-19 drug repurposing, including Zhou et al., Cell Discovery 2020 and Zhou et al., PLOS Biology 2020. The authors have to highlight new contributions of this study compared to previous studies from methodology perspectives, not call this published approach a new quantitative systems pharmacology (QSP) approach. Although the authors performed experimental validations for some predictions, current experimental assays were very preliminary and more quantitative indices to quantify inhibitory of compounds (such as IC_{50} and EC_{50} values) should be provided. Another major concern is that dose/concentration tested in Figure 6 is really high, which is out of physiological concentration.

- 1. As shown in Figure 1, the proposed QSP approach is very similar with two previous publications, Zhou et al., Cell Discovery 2020 and Zhou et al., PLOS Biology 2020. The authors should highlight new contributions of this study compared to previous Zhou's methodologies.*

We thank the Reviewer for this important comment, and for giving us the opportunity to highlight the new contributions of our study, compared to Zhou's methodologies. We agree that there are similarities between the components adopted in our QSP approach and those presented by Zhou et al., *Cell Discovery* 2020 and Zhou et al *PLoS Biology* 2020 (referred to as Zhou et al (2020b) and Zhou et al (2020c), respectively, in the manuscript). Yet, the overall workflow, the input data, selected components of the workflow, and the decision making process and criteria at critical steps differ between our approach and those in these two studies.

- **Input data:** First, the starting point (input data) in our QSP analysis is the RNA-seq data from SARS-CoV-2-infected A549, human alveolar basal epithelial cells (Blanco-Melo et al., 2020a) (Dataset 1), and those from SARS-CoV-2-infected A549 cells overexpressing ACE2 (shortly designated as A549-ACE2 cells) (Blanco-Melo et al., 2020b) (Dataset 2). In contrast, there is no SARS-CoV-2-infected cell gene expression data used as input in Zhou et al (2020b).

- **Assessment of repurposable drugs:** Consistent with our Datasets 1 and 2 that forms the basis of our QSP, the way we determined repurposable drugs was CMAP analysis of upregulated and downregulated genes in those datasets, carefully selected to comply with known effect of these genes (Fig 1C-D). This is in sharp contrast to the way repurposable drugs were identified in Zhou et al (2020b). Therein, two datasets were used: (a) the human coronaviruses (HCoV)-host interactome (assembled from four HCoVs (SARS-CoV, MERS-CoV, HCoV-229E, and HCoV-NL63), one mouse MHV, and one avian IBV); and (b) drug targets in general in the human PPI network. Repurposable drugs were identified by network proximity analysis: using the proximity of the targets of human drugs to HCoV proteins in the interactome.
- **Virus-host cell interactome:** In our case, the (already selected set of) repurposable drugs were prioritized, not selected, based on virus-host cell interactome, and we used the SARS-CoV-2 – host cell interactome reported by Gordon et al (2020). In Zhou et al., the HCoV-host cell interactomes did not include SARS-CoV-2 proteins, and interactome was used for selecting the drugs in the first place.
- **Network proximity analysis** is used in the prioritization of already selected compounds, not in their selection (as carried out by Zhou et al. 2020b). Note that this method has been introduced in 2016, and used in both studies as a component of the workflow.
- **QuartataWeb.** The use of our QuartataWeb interface was essential for determining the promiscuous targets of our selected drugs, prior to network proximity analysis. This tool was unique to our QSP analysis. This tool was used for assessing mechanisms of action and determining a representative set of compounds to be tested upon clustering the drugs based on their interaction patterns with targets.
- **Go (gene set) enrichment analysis.** Like network proximity analysis, both studies perform a gene enrichment analysis, but in different contexts. In our work, it is used at the very first step, to identify differentially expressed genes in our input datasets. In Zhou et al., it was used for validation purposes, after identification of repurposable drugs (using transcriptome data of MERS-CoV and SARS-CoV-infected cells).

So, in summary, the two QSP approaches utilize similar tools/resources (network proximity analysis, Go enrichment analysis, CMap), but (i) the overall workflow/logic is completely different (practically in opposite order, as explained above), and (ii) there are components unique to each approach (e.g. QuartataWeb identification of the first set of drug targets, selection of DEGs to build antiviral and anticytokine signatures rather than a simple signature reversal procedure, identification of mechanisms of action using QuartataWeb in our case; family-based assessment of viral proteins in Zhou et al. 2020b). We have now acknowledged these similarities and differences in a paragraph newly added to the Discussion,

Recent studies point to the utility of computational systems pharmacology approaches for identifying repurposable drugs against SARS-CoV-2 (Beck et al., 2020; Riva et al., 2020; Zhou et al, 2020b; Zhou et al 2020c; Zhou et al, 2020d; Singh et al., 2020; Gordon et al, 2020c). Of note is the work of Zhou et al (2020b)

where repurposable drugs against SARS-CoV-2 were identified by evaluating the proximity of targets of known drugs to human proteins engaged in human-CoV-host cell interactome. This type of network proximity analysis, originally introduced by Guney et al (2016), is also used here, but in a different context, mainly for prioritizing the candidate compounds/drugs that have been already identified from the DEG patterns of SARS-CoV-2 infected cells and corresponding CMap signatures. In contrast, Zhou et al (2020b) used gene set enrichment data (from MERS-CoV and SARS-CoV-infected cells) and CMap gene-drug signatures for validating their predicted drugs. Another important component unique to our analysis is the use of our interface QuartataWeb that allows for identifying drug-target associations, and for evaluating and classifying the pathways implicated in the disease modules deduced from the SARS-CoV-2-specific virus-host interactome (Gordon 2020b; 2020c) and the lung PPI network in the BioSNAP dataset (Zitnik et al., 2018) and assessing the mechanisms of action. QuartataWeb was further used to cluster the selected compounds based on their mechanisms of action and select representatives from each cluster to obtain a sufficiently diverse set for experimental testing. Thus, our study differs from that of Zhou et al (2020b) in the overall design of the computational protocol, the types of data used as input, as well as the output analyses for compound selection, prioritization, and validation, while both studies utilize state-of-the-art methods (network proximity analysis) and resources (e.g., CMap library) at different steps of the workflow.

We have now cited both papers of Zhou et al (originally, we had Zhou et al 2020b in our reference list, but not 2020c). We cited the former in the Introduction too:

We note that Zhou et al. recently proposed 16 repurposable drugs using a network proximity analysis between targets of human PPIs and host cell proteins associated with four human CoVs (SARS-CoV, MERS-CoV, HCoV-229E, and HCoV-NL63), the mouse MHV, and avian IBV, but not SARS-CoV-2 (Zhou et al., 2020b).

We are grateful to the Reviewer for bringing to our attention the 2nd paper too and helping us better explain our methodology and relevance to prior work. Please see the response to Reviewer 3 major point 1 below, for further information on the significance of the protocol in the selection of the candidate compounds.

2. Figure 1G was copied from the previous network proximity paper published by Zhou et al., Cell Discovery 2020.

We apologize for the oversight. We have replaced this diagram by our version. Nevertheless, we acknowledge in the text that this step was effectuated following the method introduced by Guney et al (2016) (who originally published a similar version of that diagram) and Zhou et al (2020b).

- 3. It is not clear how the authors selected candidate drugs/compounds for experimental validation. For example, which specific z-score cutoff from network proximity analysis are used.*
- 4. For network proximity analysis, it is unclear how the authors calculate network proximity using which human protein-protein interaction networks. The reviewer cannot find any information about human protein-protein interaction networks.*

We did not define a specific z-score. As mentioned above, network proximity method was used for prioritizing the 263 compounds identified in steps A-D shown in Fig 1. Essentially at that step, we prioritized 25 compounds for each of the four disease modules that we deduced from the SARS-CoV-2-host cell interactome reported by Gordon et al 2020c (and complemented by others' studies) upon rank-ordering the compounds based on their z-scores. Our goal was to have a sufficiently diverse set of compounds with different mechanisms of action, even though the z-score thresholds of the top 25 compounds depended on the disease module. The z-scores of the compounds were based on the proximity of their targets (determined by QuartataWeb) to proteins belonging to the disease modules. As to the human PPI network, we used the lung-specific PPI network from BioSNAP (Zitnik et al., 2018). Instead of $25 \times 4 = 100$ prioritized compounds for the four modules, we ended up with 64, as some of the compounds were associated with more than one module.

This information was presented in the original text, but we understand that some of the methodological details were apparently briefly mentioned, needed further clarification. We have generated an additional figure (current Supplementary **Figure S1**) and improved the description of the methodology, and improved several panels in Fig 1 (e.g., we showed that proximity analysis utilizes three inputs: targets of selected drugs from QuartataWeb, disease modules from SARS-CoV-2-host cell interactome, and human lung-specific PPI network from BioSNAP) to clarify these points.

Below we copied some of the paragraphs/sentences that were rewritten:

page 3

... To this aim, we used the SARS-CoV-2-host interactome (Gordon et al., 2020c) and the lung PPI network in the BioSNAP dataset (Zitnik et al., 2018) (**Figure 1F**). We first identified four *disease modules* - viral entry, viral replication and translation, cell signaling and regulation, and immune response modules in the viral-host interactome; and then, we evaluated the 'distance' of each compound from each disease module based on the proximity of the compounds' targets to the proteins belonging to the module using the lung PPI network in BioSNAP (**Figure 1G**).

page 6

Specifically, we evaluated the distance between the targets of each compound, and the proteins belonging to each module using the lung-specific PPI network from BioSNAP (Zitnik et al., 2018) and network proximity analysis (Guney et al., 2016) (see Methods).). Top-ranking 25 compounds were selected for each module (**Figure 4A** and **Table EV2**) leading to a set of 64 distinct compounds in the union of four modules (**Figure 4B**).

page 17

The compound-target interaction search engine QuartataWeb (Li et al., 2020) was used to identify targets for compounds obtained from CMap prediction, which integrates DrugBank (version 5.1.7) (Wishart et al., 2018) and STITCH (version 5) (Szklarczyk et al., 2016) databases.

page 18

In our case, we extracted the human lung protein-protein interactome from the Biomedical Network Dataset Collection BioSNAP (Zitnik et al., 2018). We defined five viral-related modules, each containing a set (S) of pre-defined proteins derived from the host proteins implicated in the SARS-Cov-2 infection (see the Results). For each compound, we have determined the set (T) of targets using QuartataWeb in the human lung PPI network. The proteins in sets S and T were connected via paths of zero or more intermediate nodes for protein. Then we evaluated the distance between these

targets and the pre-defined proteins from each viral-related module, in the human lung PPI network, as the average shortest distance path between the respective nodes s and t belonging to the sets S and T .

...

The z-scores were evaluated using the toolbox package developed by Guney et al. (2016). Note that the network proximity provides a relative measure, the absolute value of which depends on the disease and application. In the current application to four disease modules, we refrained from selecting a uniform cutoff for the z-score. Instead, we selected the top 25 compounds from each module to include a set with diverse MOAs for further analysis.

5. More details for methodology and rigor should be provided to improve reproducibility.

We have carefully examined the manuscript for descriptions of methodology and added details where needed. We described above the additional descriptions provided for the method. Every Method section and/or corresponding figure legends have information about numbers of replicates and statistical tests applied. New **Figures 6, S5, and S6** show all data points from multiple independent experiments together with appropriate measures of spread and statistical significance (e.g., SD for biological repeats and ANOVA for multiple comparisons). Dose-response curves in **Figure 6** are derived from the aggregate of up to 8 independent experiments, each performed with quadruplicate technical replicates. For matching concentrations, the data are mean \pm SD from at least three independent experiments. Where we were limited to a single experiment (**Figure 5**), replicates were incorporated into the protocol, and data subjected to single cell analysis. New violin plots in **Figure 5** show individual cell data instead of well averages, which is a much more rigorous and meaningful way to assess cellular activity than well averages that overlook heterogeneity (please see also our response to point 6). The number of data points (cells) is indicated in **Figure 5B** and statistical parameters are explained in the Figure legend.

We believe that we have described all methods in sufficient detail that others skilled in the art could repeat our studies.

6. As shown in Figure 5, the authors only shown suppression activities for predicted compounds. The authors should provide standard quantitative indices to quantify inhibitory of compounds, such as IC50 and EC50 values.

We agree that providing standard quantitative indices (e.g., IC50 or EC50 values) to quantify inhibitory effects of compounds is certainly important under most circumstances. However, our quantitative single cell analysis revealed the presence of distinct subpopulations of spike protein-expressing viral-infected cells that demonstrate differential responses to the test compounds. This intrinsic biological heterogeneity would confound the interpretation of standard inhibitory indices typically derived from averaging across an entire population. Most importantly, we feel that the acquired data and its analysis in **Figure 5** support our primary objective (of this study) of showing that the computationally predicted compounds have anti-viral activity at concentrations where overt toxicity is not appreciable (as indicated by cell loss). These results support the rationale for future studies unencumbered by BSL-3 limitations during this acute Covid-19

outbreak that will accommodate more comprehensive experimental protocols to quantify the inhibition to predict, for example, synergistic drug combinations.

7. *As shown in Figure 6, most compounds didn't show any dose-dependent responses in syncytia formation assay. In addition, dose/concentration tested in Figure 6 is really high, which are out of physiological concentration.*

Since the syncytia assay only requires BSL-2 conditions, we agree, and were able to acquire a much more extensive dataset, enabling full dose responses and the determination of EC50 values that are now presented in a new **Figure 6**. We discuss the investigational drug, linsitinib, in detail and compare its EC50 value of 25 μM to the tolerated C_{max} observed in patients, reported to be 5-10 μM . We discuss the comparison of these concentrations suggesting that they are not too disparate and additionally in this context discuss the potential for CMap to identify non canonical modes of action for a particular drug.

8. *For investigational molecules, the authors should discuss toxicity and pharmacokinetic profiles.*

We agree and as mentioned above in point #7, used the investigational drug, linsitinib, to compare clinically tolerated C_{max} values to *in vitro* EC50 values in the Discussion.

Reviewer #2:

The authors present a nice systems pharmacology method to identify drugs, targeting the host cell, with anti-SARS-CoV-2 activity and also drugs to inhibit the hyperinflammatory state in the late phase of COVID19 disease. Several of the identified drugs are experimentally validated. Identification of repurposable, host targeting drugs is an important field of the current SARS-CoV-2 related research, and the proposed pipeline theoretically can also be used in other viral infection diseases.

We thank the Reviewer for the concise summary and supportive comments.

Major comments

1. *The authors selected 36 genes from the 120 differentially expressed (DE) genes for further analysis - "Therefore, after careful evaluation, we selected 36 genes to be upregulated (Figure 2C), comprised of 26 upregulated genes associated with viral defense, and 10 downregulated genes associated with endocytic or vesicular processes." Could the authors more detailed the steps of "careful evaluation"?*

We thank the Reviewer for giving us the opportunity to provide more concrete information. We have rewritten the sentence on page 4, to replace 'careful evaluation, as

Therefore, after over-representation analysis and evaluation of the GO annotations associated with these genes as described in the Methods, we selected 36 genes to be upregulated (**Figure 2C**).

and we have rewritten the first subsection of the Methods to clarify the selection process. Below is an excerpt from that subsection, and more details are in the text:

Over-representation analysis was performed using gProfiler (Raudvere et al., 2019) with GO database (Carbon et al., 2019) for up- or down-regulated genes respectively, using Benjamini-Horchberg for multiple test correction with a threshold of 0.05. Examination of the GO Biological Process (GO-BP) and GO cellular components (GO-CC) data for up- or down-regulated genes resulted in 319 GO-BP and 13 GO-CC terms. The number of enriched upregulated terms was reduced by retaining those associated with no more than 300 genes, and not fewer than 10 overlapping genes, resulting in 16 GO terms (see *column 6* in **Appendix Table S2A**). Down-regulated terms were all kept. The enriched GO terms were organized and visualized with quickGO and classified as antiviral, proviral, or ambiguous. Those genes that defined the 'antiviral signature' were obtained by merging the up- (innate immune response) or down- (intracellular vesicle) regulated antiviral genes and excluding pro-viral (viral genome replication) components. Genes classified as proviral or ambiguous were not included in the antiviral signature.

2. *The authors write that “The A549-ACE2 cells (Dataset 2) repeatedly exhibited a more pronounced cytokine upregulation, along with IFN response insufficiency, compared to A549 cells”. The stronger response of A549-ACE2 cell line could be the consequence of the presence of ACE2 (viral receptor) on these cells, which should be discussed. Also, it is not entirely clear for me, how the authors concluded, that the gene signature of A549 cells can be used for antiviral, while the signature of A549-ACE2 cells can be used for anti-inflammatory signal generation. It would be also interesting to see how the 36 genes of antiviral signature (from A549 cells) and the 17 genes of anti-inflammatory signature (from A549-ACE2) change in the other cell lines (A549-ACE2 and A549 respectively).*

We agree that the stronger response A549-ACE2 cell line could be the consequence of the presence of ACE2 (viral receptor) on these cells, which has been described in the original paper (Blanco-Melo et al., 2020b). As suggested by the Reviewer, we analyzed how the “36 genes of antiviral signature (from A549 cells) and the 17 genes of anti-inflammatory signature (from A549-ACE2) change in the other cell lines (A549-ACE2 and A549 respectively)”. Below are the results, which we have now included as **Appendix Figure S7**. Panel **A** and **C** are the data already reported in the manuscript, i.e., the 36 genes used in the antiviral signature based on A549 cells (**A**) and 17 anticytokine-signature genes derived from A549-ACE2 cells (same as **Figs 2C** and **D**). Panels **B** and **D** display the expression levels of the same sets of genes in the other cell lines, i.e., the 36 antiviral signature genes in A549-ACE2 cells and 17 anti-cytokine signature genes in the A546-ACE2 cells. While the antiviral signature genes show similarities between the two cell types, there is a significant suppression of genes involved in the inflammatory response in A549 cells (panel **D**). This lends support to the use of A549-ACE2 cells data for detecting upregulated genes involved in inflammatory responses.

Figure S7. Comparison of the behavior of A546 and A546-ACE2 cells vis-à-vis the expression levels of the genes that have been adopted for defining antiviral and anti-cytokine signatures. **A.** The 36 genes of antiviral signature expression in A549 cells (same as **Figure 2C**). **B.** The 36 genes of antiviral signature expression in A549-ACE2 cells. **C.** The 17 genes of anti-inflammatory signature expression in A549-ACE2 cells (same as **Figure 2D**). **D.** The 17 genes of anti-inflammatory signature expression in A549 cells.

We would like to clarify that the quoted sentence was an observation (also noted earlier by Blanco-Melo et al), and we did not mean to imply that the anti-inflammatory signature must be deduced from A549-ACE2 cells but not others (nor that the anti-viral signature must be inferred from A549 cells but not others). We thought it reasonable to derive anti-inflammatory signature from A549-ACE2, as the inflammatory responses were observed to be intense and thus easy to recognize, and anti-inflammatory approaches have been suggested for SARS-CoV-2 therapy (Blanco-Melo et al., 2020b). The anti-viral signature was also based on the theoretical benefits of ‘anti-viral’ processes in viral infection. In this case, A549 cells where the observed DEGs were not partly obscured by hyperinflammatory responses, were considered. The following two sentences were added to the Discussion to clarify these points.

Cross-examination of the expression levels of the 17 anti-cytokine signature genes in A549 cells showed that most of these genes were hardly distinguished in those cells, i.e., their upregulation was specific to A549-ACE2 cells (compare panels **C** and **D** in **Appendix Figure S7**); whereas the 36 genes that define the antiviral signature exhibited a comparable expression pattern in A549-ACE2 cells (see panels **A** and **B** in **Appendix Figure S7**). These observations support the robustness of the antiviral signature on the one hand, and the utility of A549-ACE2 cells for detecting genes implicated in hyperinflammatory responses, on the other.

Minor comments

1. *The authors use not only the classical "signature-reversal" hypothesis in their work, but considered that drugs activating the host cells antiviral mechanisms can be also effective antiviral drugs. This is a biologically motivated hypothesis, which shows to be correct later in the manuscript. Several other studies investigated drug repurposing against SARS-CoV-2 using the "classical signature-reversal" hypothesis, without much experimental validation, so I found the method of the authors novel and elegant.*

Thank you very much for your appreciation of the methods used in our study.

2. *For the antiviral signature, the authors identified 100 up and 20 downregulated genes in infected A549 cells. The authors state that the upregulated genes are related to IFN response, however they state that "induction of chemokine, cytokine, and interferon types I and II were more 'muted' in SARS-CoV-2-infected A549 cells compared to those of other respiratory viruses such as influenza A and respiratory syncytial virus (Blanco-Melo et al., 2020a)". Based on which analysis did they reach this conclusion (i.e. the muted chemokine / cytokine / interferon response)?*

This phenomenon was originally described by Blanco et al (2020a), where they compared gene expression change pattern in SARS-CoV-2 models (cell lines (human alveolar adenocarcinoma (A549) cells, primary human bronchial epithelial (NHBE) cells) and ferrets), in comparison to infections of other respiratory viruses (human respiratory syncytial virus (RSV), influenza A virus (IAV)). They concluded that ‘*Compared to the response to influenza A virus and respiratory syncytial virus, SARS-CoV-2 elicits a muted response that lacks robust induction of a subset of cytokines including the Type I and Type III interferons as well as numerous chemokines.*’ However, we note that in their later more detailed study (Blanco-Melo et al., 2020b), ‘*a reduced IFN-I and -III response to SARS-CoV-2*’ of models in comparison to other viruses (IAV, human parainfluenza virus 3 (HPIV3) and RSV) is reported, while ‘*a consistent chemokine signature*’ was observed (for SARS-CoV-2). To be consistent with this updated description, we slightly modified our sentence to

Nevertheless, the induction of interferon types I and III were more 'muted' in SARS-CoV-2-infected A549 cells compared to those of other respiratory viruses such as influenza A and respiratory syncytial virus (Blanco-Melo et al., 2020b).

3. *In Figure 5B the authors show the results of the viral infection assay. I think a violin or swarm plot would be more suitable for displaying the results than the current boxplot + distribution plot.*

We agree. Particularly for a broader readership that may be more familiar with the violin plots, we made the requested change in **Figure 5B**.

4. *I generally felt the antiviral part of the study very convincing, both from the side of systems pharmacology method and the experimental validation. On the other side the anti-hyperinflammatory part is weaker in my opinion. At first, the authors also used the virus-host protein interactions and a lung specific PPI network (if I understand correctly) to prioritise this list. I think the COVID19 related hyperinflammatory state is more related to different immunocytes than the originally infected lung cells. Despite these (theoretical)*

considerations, the identified drugs look promising eg. glucocorticoid agonists and TNF α inhibitors. However, I was not able to find glucocorticoid agonists and TNF α inhibitors in the corresponding Table4. Also, some (at least literature-based) validation of these drugs would be also suitable.

Thank you for these insightful comments. We have used the virus-host protein interactions and a lung specific PPI network for prioritizing the compounds in the antiviral part. As to the anti-cytokine part, we did not do this type of network proximity calculation. Instead, we directly analyzed the 275 compounds deduced from CMap signature reversal search (**Table S3B**) to cluster 163 of them for which there were available target data into groups based on their drug-target interaction patterns as computed by QuartataWeb (**Table S4B**). The remaining 112 were manually examined to prioritize three, resulting in the **Table 4**. We have edited **Table 4** to clarify these two groups, and how they were obtained. We apologize for the confusion regarding the properties of potential anti-inflammatory drugs included in that Table. We have edited and expanded the discussion on the identified compounds in the following paragraph:

Table 4 contains 15 FDA approved drugs and 12 compounds under investigation. Of note, two of the compounds under investigation (JAK3-Inhibitor-II and AZD-8055; in *boldface*) also belong to the 64 top-ranking compounds based on Dataset 1; and one, mepacrine/quinacrine, is listed in the Excelra COVID-19 drug repurposing database (Excelra, 2020). Another investigational drug in the list, PCA4248, is a platelet activating factor (PAF) receptor antagonist (Fernandez-Gallardo et al., 1990), and its utility against Covid-19 (e.g., for preventing coagulation or blood clots) is to be explored, as well as those of the two His receptor antagonists azelastine and chlorphenamine, identified here. Recent study draws attention to the possible repurposing of PAF receptor antagonists and His receptor antagonists against hyperinflammation and microthromboses in COVID-19 patients (Demopoulos et al, 2020).

Among approved drugs, pirfenidone, is known to inhibit furin (Burghardt et al., 2007), a human protease involved in the cleavage of the viral spike glycoprotein into S1 and S2 subunits (like TMPRSS2). Spike cleavage is essential to activate the S1 fusion trimer for viral entry. Pirfenidone combined with melatonin has been pointed out to be a promising therapy for reducing cytokine storm in COVID-19 patients (Artigas et al, 2020). Finally, **Table 4** also contains two approved cyclooxygenase inhibitors, oxaprozin and dexketoprofen, known as non-steroidal anti-inflammatory drugs (NSAIDs) (Miller, 1992; Moore and Barden 2008).

Reviewer #3:

The authors present results from an original system pharmacology method aiming at identifying small molecules and drug repositioning opportunities for treating SARS-CoV-2 infected patients.

The presented method is based on the nowadays famous CMap database and relative signature matching approach successfully used in previous works based on a 'signature reversion principle'. In most of these works a transcriptional signature characterizing a disease phenotype to be rescued is first identify then used to find drugs exerting an 'inverse' effect on cellular transcriptional program, i.e., up-regulating genes that are down-regulating in the disease signature and vice-versa. The hypothesis is that reverting the disease signature might revert

(thus rescue) the disease phenotype itself. This approach has been proven effective in a number of work and formally demonstrated at least for metabolic disorders.

Here the authors go one step further, designing a sort of 'rationale connectivity mapping' approach where the transcriptional signature summarizing the phenotype to be rescued or (as in this case) just modulated is generated from two condition specific sets of differentially expressed genes (from publicly available RNA-seq data) are assembled, combined and then refined to identify a component that should be reverted (the anti-viral component) and another one that should be enhanced (to reduce inflammatory host-response). This is accomplished via a knowledge based (or semi-supervised) approach based on the observation and selection of relevant enriched GO categories).

By using these signatures (assembled from publicly available data) against the CMap, the authors are able to identify large sets of hit compounds that are further prioritized using a network proximity-based approach (using a recently published SARS-CoV-2 specific network). Shortlisted hits are finally successfully experimentally validated.

Briefly this is a nice piece of work that extends an established computational paradigm in an innovative and original way and whose results are robustly validated and might have a great impact. Few major points should be addressed before further considering this manuscript for publication in Molecular Systems Biology.

We thank the reviewer for the insightful summary and for the overall support and appreciation.

Major points:

- 1. Whereas the idea of a 'rationale' or semi-supervised connectivity mapping is valid and justified, the GO:term guided selection of the genes to be included in the final query to the CMap appears to overwhelm the transcriptional data, with (for example) a mixture of up/down regulated genes included in the anti-viral component of the final signature. What would happen if starting just from intersecting all relevant GO:categories to compose the set of genes to be up- down-regulated by the CMap compounds? The author should convincingly show that a pure knowledge based approach just based on GO:categories or signatures from the MsigDB database would lead to different results and set of compounds outputted when querying the CMap and that there is indeed value in combining such approach with the initial enrichment analysis of the genes differentially expressed upon viral infection.*

GO:categories *per se* are valuable, however, these annotations are derived from previous knowledge of molecular pathways, known diseases or experiments, while it remains to be proven that they are also capable of describing the expression changes specifically induced by the newly emerged SARS-CoV-2 infection. This was the motivation underlying the utilization of the differentially expressed genes from SARS-CoV-2 infected models. Examination of the GO terms associated with the DEGs (36 antiviral and 17 anti-cytokine genes) shows that the DEGs represent only a subset of the proteins associated with different GO terms, which indicates that only certain, but not all genes participating in a given 'GO' pathway are affected by SARS-CoV-2 infection. This type of behavior specific to SARS-CoV-2 infection, might be missed if compounds were derived directly based on GO:categories or on signatures from the MsigDB database. The analysis below provides support to this concept.

To address this specific question from the reviewer, we performed the following analysis (for A549 cells). We considered three additional lists of CMap-predicted compounds (CMap score > 90) from related GO terms, without using any knowledge on DEGs:

- 1) an antiviral list of 32 genes, from the intersection of GO:0034341 (response to interferon gamma) and GO:0051607 (defense response to virus), as ‘to be upregulated list’ in CMap query. To this aim, we used the same parameters as those in our original query (for identifying the 36-gene antiviral signature). The list of these new compounds is called ‘GO:antiviral’.
- 2) a proviral list of 129 genes, from GO:0019079 (viral genome replication), as ‘to be upregulated list’ in CMap query, again using the same parameters as those used to generate the original list (from 36 gene antiviral signature). This list is called ‘GO: proviral reversal’.
- 3) An antiviral list of 22 genes (from the aforementioned antiviral 32 genes in 1), and a proviral list of 81 genes (from the aforementioned proviral 129 genes in 2), excluding genes if they are either in GO:0034341 (response to interferon gamma) or GO:0051607 (defense response to virus)). The Venn plot below shows the overlaps between the three GO terms. The antiviral list of 22 gene was used as ‘to be upregulated list’ with simultaneous input of the proviral list of 81 genes as ‘to be downregulated list’ in CMap query. The new compound list was named ‘GO:antiviral and proviral reversal’.

The overlap of selected compounds from the original list (utilizing both DEG information and GO) and these three additional lists (using GO only) is shown in the Venn plot below. The list can be found in [overlap_compounds.xlsx](https://pitt.box.com/s/k71zloa0cv88vc4o59l99c2ooj7jozbx).

This analysis shows that CMap-based compounds do show differences with versus without DEG information, while validity of each predicted list of compounds require wet lab validation for claiming their true effectiveness.

- Components 3 and 4 of figure 1A seem to refer only to the genes to be upregulated. Why there isn't an equivalent couple of visuals for the 17 genes to be downregulated?

Thank you for the excellent point. We have revised **Figure 1** and included the 17 genes to be downregulated.

- It is not immediately clear if the two components of the signature were used as a single query or into two separate instances. This should be explicitly mentioned. The authors report two sets of compounds outputted by their query: a set of potentially anti-viral compound and a set of potentially anti-cytokine compounds, hinting that there were two queries, but they also wrote 'our query/input signature'. This is very confusing.

We are so sorry about causing confusion. In our study, we have obtained two separate gene signatures that were used in two separate instances, and we obtained two separate sets of output - potential antiviral-compounds output and potential anti-cytokine compounds - corresponding to the respective SARS-CoV-2 infected A549 and A549-ACE2 cells' transcriptomes. We corrected the phrase that caused confusion as

“the query/input signature, repeated separately for the antiviral and anti-cytokine signatures”.

- The benchmark output against Excelra, QuartataWeb etc. is important and should be summarized somewhere in a figure panel.

Thank you for the suggestion. We have created a new figure, **Appendix Figure S1**, for better describing the overall protocol. The steps involving the comparison with Excelra DB and the utilization of QuartataWeb are displayed in the figure.

- There is a lack of statistical assessment of the target proximity to the disease relevant modules identified in the SARS-CoV-2 specific interaction network. What is the expected distance of randomly selected nodes to any of the modules? The author should use a less arbitrary threshold possibly based on this assessment to further prioritize compounds instead of just picking the top ranking 25 compounds etc.

The z-scores themselves are evaluated based on a rigorous statistical analysis. We have expanded the section in the Methods to better explain how we are evaluating a statistical score based on the expected behavior of all distances whose distributions have been evaluated, not the distances themselves. We copied below the corresponding section (from pages 17-18):

The proteins in sets S and T were connected via paths of zero or more intermediate nodes for protein. Then we evaluated the distance between these targets and the pre-defined proteins from each viral-related module, in the human lung PPI network, as the average shortest distance path between the respective nodes s and t belonging to the sets S and T, as

$$d(S,T) = \frac{1}{\|T\|} \sum_{t \in T} \min_{s \in S} d(s,t)$$

Then a reference distance distribution was constructed, corresponding to the expected distance between the disease module proteins and a randomly selected groups of proteins in the network, with the same size and degree distribution as the original disease proteins and drug targets in the network. This procedure was repeated 1,000 times, and the mean and standard deviation of the reference distance distribution were used to calculate a z-score

6. *The discussions section is long and quite boring. I do not believe that it is necessary to list all those details for the identified compounds. The authors should limit to discussing their approach and findings and potential implications.*

Thank you for the suggestion. We thought it would be useful to the readers to have some information on the selected compounds and present our findings in the context of the observations and findings from other labs. But given the criticism of the Reviewer, we have shortened the section on the details of the identified compounds. We think the material that we kept in the Discussion may be important to some readers to streamline future work.

Minor points:

1. *In the introduction the authors claim that "Many compounds under clinical trials against SARS-CoV-2 are repurposable drugs". Given that the compounds the authors are referring to are still under clinical investigation, this sentence should be amended as "Many compounds under clinical trials against SARS-CoV-2 are potentially repurposable drugs".*

Thank you. We amended the sentence as recommended.

2. *'Quantitative systems pharmacology' is used 4 times through all the manuscript. Does this really require introducing an acronym? which is confused and also used in a section title (which should be generally avoided).*

We removed the acronym, as suggested.

3. *A CMap based drug signature refinement approach aiming at improving drug repositioning predictions and moving in the same alley of the approach presented in this manuscript has been introduced recently and could be cited (PMID: 26452147).*

We thank the Reviewer 3 for bringing to our attention this study, which we have now cited in the Introduction.

Thank you for sending us your revised manuscript. We have now heard back from the three reviewers who were asked to evaluate your study. As you will see the reviewers are overall satisfied with the modifications made and think that the study is now suitable for publication.

Before we can formally accept your manuscript, we would ask you to address the following issues:

REFEREE REPORTS

Reviewer #1:

The authors have addressed my major concerns in previous review. The authors argued that Zhou et al., Cell Discover 2020, only looked other virus, not SARS-CoV-2. Yet, the second study, Zhou et al., PLOS Biol 2020, systematically inspected anti-SARS-CoV-2 repurposable drugs using network proximity and GSEA approaches as the authors used in this study. As shown in Figure 6, the overall EC50 value of tested drugs is poor; yet, nafamostat has been widely studied in various antiviral studies. Overall, the reviewer support the revised manuscript can be published based on the in pressing need of repurposable drugs for COVID-19 and systematic computational and experimental validations done by this study although with limited novelty of proposed computational approaches.

Reviewer #2:

We thank the authors for addressing our comments and we are satisfied with their responses. From what we can tell they have also addressed well the other reviewers.

Reviewer #3:

The authors performed an excellent job while addressing mine and other reviewers' comments. I recommend this manuscript for publications in Molecular Systems Biology

The authors have made all requested editorial changes.

Thank you again for sending us your revised manuscript. We are now satisfied with the modifications made and I am pleased to inform you that your paper has been accepted for publication.

Corresponding Author Name: Ivet Bahar

Manuscript Number: MSB-2021-10239